# Photoinduced hydrogen dissociation in thymine predicted by coupled cluster theory

Eirik F. Kjønstad [1,2,3] ✉, O. Jonathan Fajen [1,2], Alexander C. Paul [3],
Sara Angelico [3], Dennis Mayer[4], Markus Gühr [4,5], Thomas J. A. Wolf [1],
Todd J. Martínez [1,2] ✉ & Henrik Koch [3] ✉

The fate of thymine upon excitation by ultraviolet radiation has been the subject of intense debate. Today, it is widely believed that its ultrafast excited state gas phase decay stems from a radiationless transition from the bright $\pi\pi^*$ state to a dark $n\pi^*$ state. However, conflicting theoretical predictions have made the experimental data difficult to interpret. Here we simulate the early gas phase ultrafast dynamics in thymine at the highest level of theory to date. This is made possible by performing wavepacket dynamics with a recently developed coupled cluster method. Our simulation confirms an ultrafast $\pi\pi^*$ to $n\pi^*$ transition ($\tau = 41 \pm 14$ fs). Furthermore, the predicted oxygen-edge X-ray absorption spectra agree quantitatively with experiment. We also predict an as-yet uncharacterized $\pi\sigma^*$ channel that leads to hydrogen dissociation at one of the two N-H bonds. Similar behavior has been identified in other hetero-aromatic compounds, including adenine, and several authors have speculated that a similar pathway may exist in thymine. However, this was never confirmed theoretically or experimentally. This prediction calls for renewed efforts to experimentally identify or exclude the presence of this channel.

Thymine, like other nucleobases, undergoes ultrafast radiationless relaxation back to the ground state after being excited by ultraviolet radiation. This property has been tied to the resilience of genetic material against light-induced damage[1]. However, the exact mechanism of this decay is not fully understood and has been a subject of debate for several decades. Gas phase experiments have identified at least two excited state decay channels, one with a lifetime on the order of ≲100 fs, and one considerably longer, on the order of several ps[2–5]. Yet, the underlying mechanisms have been challenging to discern, with proposed explanations necessarily relying on simulations of the molecular dynamics. These simulations, in turn, introduce approximations with errors that are difficult to control. Different theoretical methods, and in particular different electronic structure methods, have therefore produced different explanations, and a consensus has yet to emerge.

Most reported simulations implicate two low-lying excited states in the relaxation: a dark $n\pi^*$ state ($S_1$) and the bright $\pi\pi^*$ state ($S_2$) into which the system is initially excited. The interplay of a bright $\pi\pi^*$ and a dark $n\pi^*$ state is central to the excited-state relaxation of a wide variety of other chromophores, including nucleobases such as uracil and adenine, although lifetimes and branching ratios differ. In the case of thymine, several simulations predict a $\pi\pi^*$ trapping channel in which the initial excitation to the $\pi\pi^*$ state is rapidly followed by relaxation into a minimum on the $\pi\pi^*$ surface, where the wavepacket is trapped for tens or hundreds of fs[6–12]. These simulations disagree, however, on the amount of $\pi\pi^*$ trapping, as well as the timescale and nature of the subsequent processes, with proposed mechanisms including $\pi\pi^*$ to $n\pi^*$ relaxation[11] and direct $\pi\pi^*$ relaxation to the ground state[8]. Some of these studies indicate that the amount of $\pi\pi^*$ trapping is reduced by improving the description of dynamical correlation (instantaneous

[1]Department of Chemistry, Stanford University, Stanford, CA, USA. [2]Stanford PULSE Institute, SLAC National Accelerator Laboratory, Menlo Park, CA, USA. [3]Department of Chemistry, Norwegian University of Science and Technology, Trondheim, Norway. [4]Deutsches Elektronen-Synchrotron DESY, Hamburg, Germany. [5]Institute of Physical Chemistry, University of Hamburg, Hamburg, Germany. ✉e-mail: eirik.kjonstad@ntnu.no; todd.martinez@stanford.edu; henrik.koch@ntnu.no

electron-electron interactions)[8,11], a pattern that has also been found in the closely related nucleobase uracil[13]. In line with this, an early density functional theory (DFT) study found a significant $n\pi^*$ population within the first 50 fs[14] and a more recent mixed-reference spin-flip DFT study also found rapid barrier-less $\pi\pi^*/n\pi^*$ transfer ($\tau$ = 30 fs), including subsequent $n\pi^*$ trapping[15].

Experimental evidence has implicated the $n\pi^*$ state in the early dynamics. Indeed, by determining the gas phase oxygen-edge time-resolved X-ray absorption spectrum, Wolf et al.[16] found a fast component ($\tau$ = 60 ± 30 fs) which was attributed to population of the $n\pi^*$ state. This was further corroborated in a recent time-resolved photo-electron spectrum reported by Miura et al.[17] ($\tau$ = 39 ± 1 fs). Thus, the wavepacket appears to already transfer some of its population to the $n\pi^*$ state within the first 100 fs. Furthermore, Wolf et al. found that the $n\pi^*$ signature lasts for several ps, revealing a second relaxation mechanism. After passing through the $\pi\pi^*/n\pi^*$ conical intersection, parts of the wavepacket appears to get trapped in a minimum on the $n\pi^*$ surface. On the timescale of a few ps, there is evidence that thymine further undergoes intersystem crossing from the $n\pi^*$ singlet to a $\pi\pi^*$ triplet[18]. While experiments have thus shown that the $n\pi^*$ state is involved in the early sub-100 fs dynamics, it remains an open question whether or not there is some trapping in the $\pi\pi^*$ state[19,20]. More accurate simulations of the dynamics are therefore essential to unravel the precise relaxation mechanisms in thymine.

Here we present the highest-level wavepacket simulation on the early dynamics of gas phase thymine to date. To the best of our knowledge, this is also the simulation with the highest level of single-reference electronic structure theory performed on a molecular system of this size. Thanks to recent developments[21–26], we were able to describe the electronic structure with the highly accurate coupled cluster singles and doubles (CCSD)[27] method in its equation of motion formulation for excited states (EOM-CCSD)[28]. To the best of our knowledge, this method has not been applied in nonadiabatic dynamics before. Coupled cluster (CC) theory is well-known for effectively capturing dynamical correlation, but it has been widely regarded as unsuited for excited state dynamics due to the presence of numerical artifacts at conical intersections[21,29–31]. Recent work has shown that these problems can be removed with similarity constrained CC (SCC) theory[22,23,26]. Here, we apply the SCC with singles and doubles (EOM-SCCSD) method to simulate the first 100 fs after photoexcitation using ab initio multiple spawning (AIMS)[32,33], showing that it is possible to simulate nonadiabatic dynamics with a coupled cluster method that correctly describes conical intersections. This demonstrates that coupled cluster theory is a viable electronic structure method for simulating a range of photochemical processes.

## Results and discussion

The main photochemical pathway in our simulation is illustrated in Fig. 1. First, our simulation confirms an ultrafast $\pi\pi^*$ to $n\pi^*$ conversion. We find no significant $\pi\pi^*$ trapping and no direct $\pi\pi^*$ relaxation to the ground state in the first 100 fs. The population transfer from $\pi\pi^*$ to $n\pi^*$ is followed by trapping in an $n\pi^*$ minimum on $S_1$. This $n\pi^*$ trapping channel constitutes the main (87%) photochemical channel of the simulation. In addition to this ultrafast $\pi\pi^*$ to $n\pi^*$ conversion, our simulation predicts a minor $\pi\sigma^*$-mediated N-H dissociation channel (13%).

The $n\pi^*$ trapping channel can be experimentally identified from the gas phase time-resolved oxygen-edge X-ray absorption spectrum. In Fig. 2, we compare our simulated spectrum with the experimental X-ray absorption spectrum reported by Wolf et al.[16]. The most striking feature in these spectra is a bright signal, at around 526 eV, that grows in intensity in the first 50–70 fs. The theoretical and experimental spectra are in agreement for this feature (see panels C and D), which we find to be due to population transfer from $\pi\pi^*$ to $n\pi^*$, in agreement with the mechanism proposed in ref. 16. By analyzing the simulation

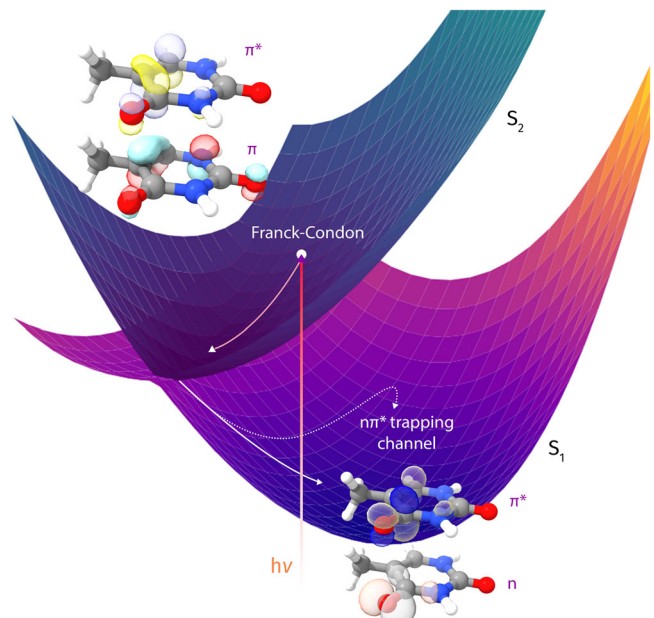

**Fig. 1 | Main photochemical pathway in the dynamics simulation.** Potential energy surfaces are shown for the first and the second excited states ($S_1$ and $S_2$). Following photoexcitation to the bright $\pi\pi^*$ state ($S_2$), the simulation predicts two channels. The main channel is the $n\pi^*$ trapping channel. Here, the wavepacket passes through the $S_1/S_2$ intersection and heads toward an $n\pi^*$ minimum on the $S_1$ surface. This minimum is reached in two ways, either by heading to the minimum directly (solid line) or by reaching it indirectly through a $\pi\pi^*$ region on $S_1$ (dashed line).

data, we find a characteristic time of $\tau$ = 41 ± 14 fs for this conversion (see Suppl. Note 1), which fits well with experimental estimates (60 ± 30 fs[16] and 39 ± 1 fs[17]). In the simulated spectrum, we also find weak but visible features that are associated with the initial $\pi\pi^*$ dynamics: in the first 20 fs, there are broad and diffuse features at around 526–528 eV and at 534 eV, where the 534 eV feature is partially hidden by the ground state bleach (see Fig. 2B). These features reflect the rapid movement of the wavepacket away from the Franck-Condon region and towards the $S_1/S_2$ conical intersection. Moreover, we find oscillations in the signal at 526 eV associated with dynamics on the $n\pi^*$ state. These oscillations have a period of about 20 fs and an amplitude of about 1 eV (see panel B). The limited 100 fs FWHM time resolution of the experiment[16] washes these oscillations out (see Fig. 2C), but we predict that they would be observed with improved time resolution. The oscillations are similarly washed out when we apply the same time broadening to the simulated spectrum (see Fig. 2D).

The $\pi\pi^*/n\pi^*$ conversion time of $\tau$ = 41 ± 14 fs was determined by analyzing the growth of the 526 eV signal in the simulated spectrum. This time constant is consistent with the rate of $\pi\pi^*/n\pi^*$ conversion in the simulated dynamics, that is, from the observed change in electronic character from $\pi\pi^*$ to $n\pi^*$. We find a rapid adiabatic population transfer from $S_2$ to $S_1$ ($\tau$ = 17 ± 1 fs) in our simulation. However, when the adiabatic states are decomposed into their diabatic components, and in particular into their $\pi\pi^*$ and $n\pi^*$ components, we see that the growth in the $n\pi^*$ character ($\tau$ = 37 ± 9 fs) is in close agreement with the time constant determined from the simulated spectrum ($\tau$ = 41 ± 14 fs). This shows that the 526 eV signal in the spectrum is due to the electronic $n\pi^*$ character.

To better understand the dynamics, and in particular the $n\pi^*$ trapping channel, we identify stationary points on the $S_1$ and $S_2$ surfaces in the vicinity of the Franck-Condon region. In agreement with previous calculations, we find an $n\pi^*$ minimum on the $S_1$ surface at an extended $C_4-O_8$ bond length[11,16]. This extension is due to the

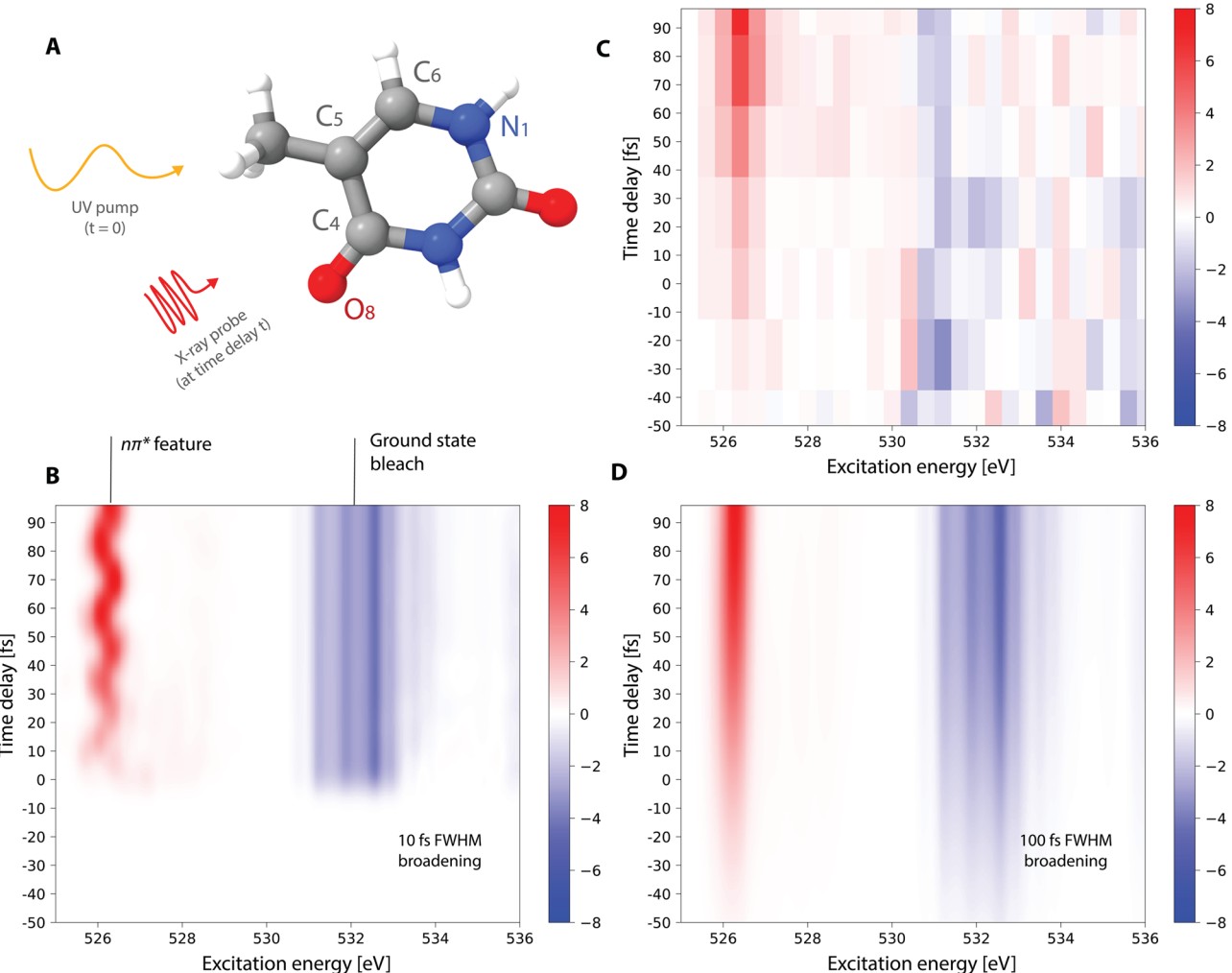

**Fig. 2 | Thymine oxygen-edge X-ray absorption spectrum.** Panel **A** illustrates the pump-probe scheme for thymine, with an ultraviolet (UV) pump and an X-ray probe, and the numbering used for the important atoms in the dynamics. Panels **B**–**D** show oxygen-edge X-ray absorption spectra for thymine, simulated (**B** and **D**) and experimental (**C**)[16]. Color bars express the intensities in arbitrary units. Simulated spectra were computed using CC3/cc-pVDZ transition strengths and AIMS dynamics with CCSD/cc-pVDZ and SCCSD/cc-pVDZ, together with Gaussian broadening with a full width at half maximum (FWHM) of 0.3 eV/10 fs (panel **B**) and 0.3 eV/100 fs (panel **D**). For more details, see Suppl. Note 2. The most prominent feature in the spectrum is a bright signal at 526 eV, which is due to the $n\pi^*$ state and reflects the rapid $\pi\pi^*/n\pi^*$ internal conversion after photoexcitation to the $\pi\pi^*$ state (at time $t = 0$). The oscillations in the $n\pi^*$ feature (see panel **B**) are washed out by a larger time broadening (see panel **D**), reflecting the time resolution of the experiment (see panel **C**).

anti-bonding character of the $\pi^*$ orbital along the bond. We also find an $S_1/S_2$ minimum energy conical intersection (MECI) that can be reached from the Franck-Condon region through $C_5$-$C_6$ elongation. Close to such intersections, CCSD exhibits numerical artifacts (unphysical complex energies and distorted potential energy surfaces) which can be removed with SCCSD. Figure 3 details a simulated initial condition that encounters such artifacts and shows how the issue is averted with the SCCSD method.

By analyzing stationary points on $S_1$ and $S_2$, Wolf et al.[16] suggested that the excited state decay of thymine follows a two-step process in the $C_5$-$C_6$ and $C_4$-$O_8$ coordinates (see Fig. 2A for atom labeling): following photoexcitation, the $C_5$–$C_6$ bond is first elongated, and along this stretching coordinate, the $S_1/S_2$ intersection seam is accessible; then, after interconversion to the $S_1$ state, the $C_4$-$O_8$ bond elongates as the wavepacket heads towards the $n\pi^*$ minimum on $S_1$. This picture is borne out by our dynamics simulation. In Fig. 4, we show the time evolution of the nuclear density in the $C_5$–$C_6$ and $C_4$–$O_8$ bond coordinates, and we indeed see this two-step process unfolding in real time. For an initial condition simulated using an extended basis set (aug-cc-pVDZ), we also find that the initial dynamics follows the same path from the Franck-Condon region to the $S_1/S_2$ intersection region (see Suppl. Note 3).

The most intriguing result in the dynamics simulation is the presence of a hydrogen dissociation channel (13%). This finding comes as a surprise because no such dissociative channel has been previously reported. In fact, total kinetic energy release (TKER) spectra across a broad range of wavelengths, 270–230 nm, showed no signature of ultrafast N-H dissociation[34], and recent experimental and theoretical investigations (see, e.g., refs. 11,15,18,19) have not invoked this pathway to explain the molecular dynamics. However, a similar decay mechanism has been proposed in adenine, where a dark $\pi\sigma^*$ state is accessed through interconversion from the $\pi\pi^*$ state[35,36]. More generally, the importance of these low-lying dissociative $\pi\sigma^*$ states are well-recognized in a wide variety of heteroaromatic systems[37–40]. In adenine, the $\pi\sigma^*$ state is dissociative along the N-H bond coordinate and intersects with the ground state ($S_0$) at larger N-H distances. Following excitation to the $\pi\pi^*$ state, the wavepacket may reach this $\pi\pi^*/\pi\sigma^*$ conical intersection within 20 fs. Once on the repulsive $\pi\sigma^*$

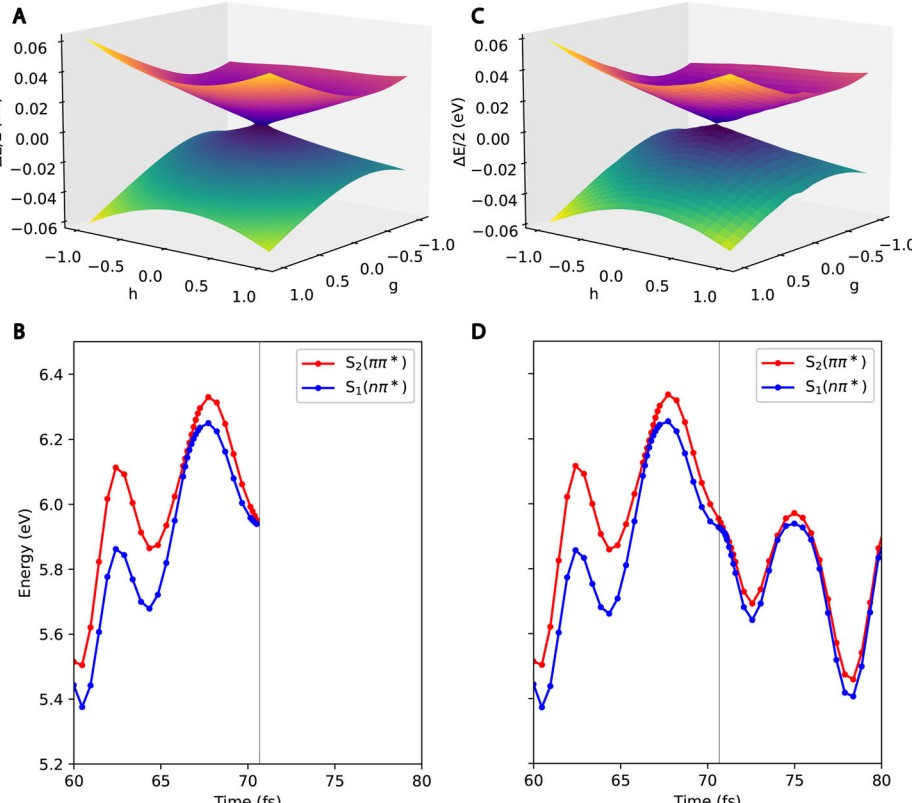

**Fig. 3 | An example of a numerical artifact encountered during the dynamics with EOM-CCSD (panels A and B) and their correction with EOM-SCCSD (panels C and D).** Panels **A** and **C** show a branching or *gh* plane for a conical intersection encountered during the simulation of one of the initial conditions. The positive and negative energy difference between the first and second excited states ($S_1$ and $S_2$) divided by two ($\Delta E/2$) is shown. The *g* and *h* vectors are energy-rescaled and the displacements *g* and *h* are in arbitrary units. For CCSD (**A**), there is an ellipse of degeneracy in the *gh* plane with unphysical complex-valued energies in the interior of the elliptical boundary; for SCCSD (**C**), we instead see a single point of degeneracy and no unphysical energies. In CCSD simulations, the wavepacket may

approach the intersection too closely and end up in the region with complex-valued energies (the interior of the ellipse in the *gh* plane). Whenever this happens, we re-run the simulation with SCCSD. Panels **B** and **D** show the corresponding potential energy curves for the center of a nuclear trajectory basis function with CCSD (**B**) and the same trajectory basis function with SCCSD (**D**). At 71 fs, the CCSD simulation enters the complex-valued region and is terminated (**B**). The SCCSD simulation, on the other hand, does not encounter any problems (**D**). The expansion point used in the branching plane calculation is the geometry with the smallest $\Delta E$ (as given by SCCSD) in the nonadiabatic event at 71 fs.

state, N-H dissociation proceeds rapidly, resulting in a characteristically anisotropic TKER spectrum of H atom fragments[41]. Given the finding of a similar $\pi\sigma^*$ channel in our simulation on gas phase thymine, it may be that certain regions of the $\pi\pi^*$ surface—in particular, regions accessed by a highly excited N-H stretching mode—lead to rapid relaxation through an intersection with a $\pi\sigma^*$ state, followed by hydrogen dissociation at the N-H bond.

Figure 5 characterizes this channel in terms of the normal mode associated with the $N_1$-H stretch (see panel A). When thymine is displaced along the mode away from the equilibrium geometry ($Q = 0$), a dissociative $\pi\sigma^*$ state comes down and intersects with $n\pi^*$ and $\pi\pi^*$ states at around $Q = 0.4$ (see panel B), which corresponds to an $N_1$-H bond length of around 1.3 Å. At longer $N_1$-H bond lengths, the character of $S_1$ is $\pi\sigma^*$ (see panel C). The two dissociating initial conditions both have a short initial $N_1$-H bond length, and they both transfer population to $S_1$ at long $N_1$-H bond lengths (see panel D). The simplest explanation for the dissociative channel, therefore, is that initial conditions with short $N_1$-H bonds experience a rapid extension of the bond in the initial phase of the nuclear dynamics, allowing them to access the $\pi\sigma^*$ state. This suggests that initial conditions with shorter $N_1$-H bonds at $t = 0$ would be more likely to dissociate. This indeed appears to be the case: out of 17 additional initial conditions specifically selected to have $N_1$-H bond lengths shorter than 0.9 Å, we find that 7 of the conditions (41%) have access to parts of the potential energy surfaces with $\pi\sigma^*$ character (see Suppl. Note 4).

In the dynamics simulation, only 2 out of 16 initial conditions (13%) lead to hydrogen dissociation at the $N_1$-H bond. Given the limited number of initial conditions, it is difficult to estimate how common this pathway is. Inspection of the natural transition orbitals (NTOs) of one of the conditions shows how the dissociation happens (see Suppl. Note 4). The other initial condition has similar behavior. Already at 2.5 fs, $S_3$ is mainly of $\pi\sigma^*$ character, while $S_2$ is still of $\pi\pi^*$ character and $S_1$ of $n\pi^*$ character. As the wavepacket moves along the $N_1$-H stretching coordinate, the $\pi\sigma^*$ state is stabilized, ultimately falling below the $\pi\pi^*$ and $n\pi^*$ states. The $S_3/S_2$ gap is about 0.5 eV at 2.4 fs, but by 3.4 fs, this gap has decreased to 0.1 eV. At the same time, $S_3$ is of mainly $\pi\sigma^*$ character at 2.4 fs, but by 4.0 fs, the character of $S_2$ and $S_3$ has flipped. Eventually, at around 4.1 fs, there is an intersection between $S_1$ and $S_2$, where most of the population is transferred to $S_1$. Moving away from the intersection, $S_1$ is dominated by $\pi\sigma^*$ character. $N_1$-H dissociation occurs rapidly over the next 15 fs, eventually reaching a ground state intersection at an extended $N_1$-H bond length (greater than 2.0 Å).

Additional calculations support the predicted $\pi\sigma^*$ pathway. Both of the dissociative initial conditions display identical behavior when the $S_3$ state is included in the dynamics simulation, showing that its inclusion does not suppress the channel (see Suppl. Note 6). Moreover, while the $\pi\sigma^*$ state has Rydberg character in the Franck-Condon region, this character decreases as the $N_1$-H bond extends and the state becomes involved in the dynamics. Potential energy curves along the

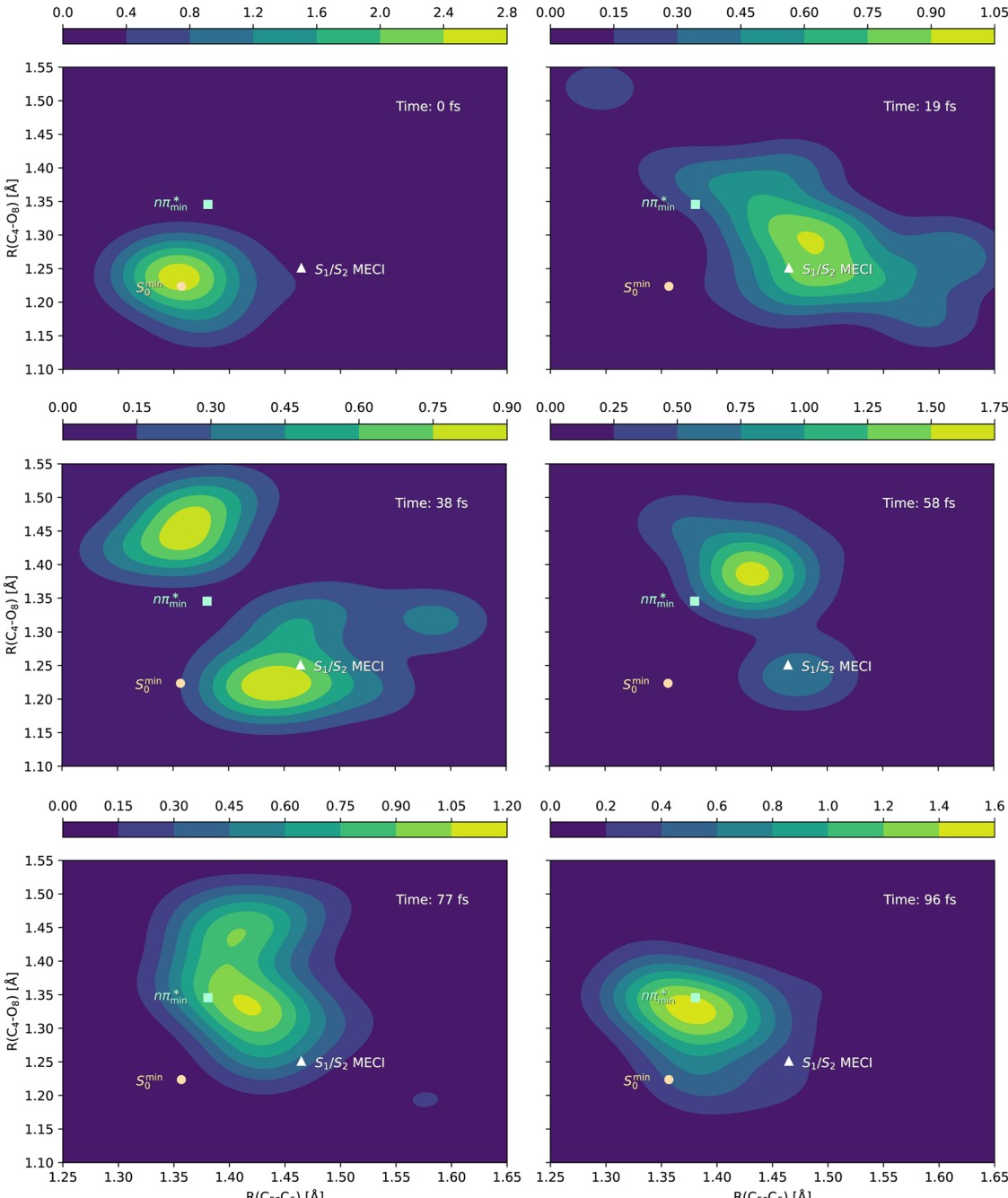

**Fig. 4 | Total nuclear density snapshots in the $C_5$-$C_6$ and $C_4$-$O_8$ coordinates.** Three important stationary points are shown: the ground state minimum $S_0^{min}$, the minimum energy conical intersection $S_1/S_2$ MECI, and the $S_1$ minimum $n\pi^*_{min}$. The wavepacket quickly moves away from the Franck-Condon region ($S_0^{min}$) and towards the minimum-energy conical intersection, where it starts transferring population to the $n\pi^*$ surface (19 fs), eventually causing the wavepacket to split (38 fs). At longer times, after almost all of its population has transferred to the lower surface (58 fs), the wavepacket settles in the vicinity of a minimum on the $n\pi^*$ surface (77 and 96 fs).

$N_1$-H bond with a diffuse basis set (aug-cc-pVDZ) and with a higher-order correlation treatment (coupled cluster with perturbative triples, CC3[42]) suggest that the pathway is still present with more accurate treatments (see Suppl. Note 7).

Current experimental data does not allow us to verify or exclude the existence of the dissociative channel. We do not find any clear signature of the $\pi\sigma^*$ channel in our simulated spectrum (see Fig. 2). However, the channel appears to be small (~13%), and an oxygen-edge spectrum is not expected to be highly sensitive to changes at the $N_1$-H bond. A nitrogen-edge spectrum, on the other hand, should be sensitive to these changes, as indicated by

calculations at selected geometries (see Suppl. Note 8). However, no experimental time-resolved nitrogen-edge spectrum has been reported.

Some experimental data have appeared to disconfirm the existence of a dissociative $\pi\sigma^*$ channel, as reported TKER spectra showed only smooth, isotropic H-atom kinetic energy, indicating no involvement of ultrafast, $\pi\sigma^*$-mediated N-H dissociation[34]. However, these spectra only scanned over excitation wavelengths from 270–230 nm, and dissociation might become more common at shorter wavelengths. For adenine, the TKER spectra for wavelengths from 280–234 nm also show smooth, isotropic H-atom kinetic energies. Only for wavelengths

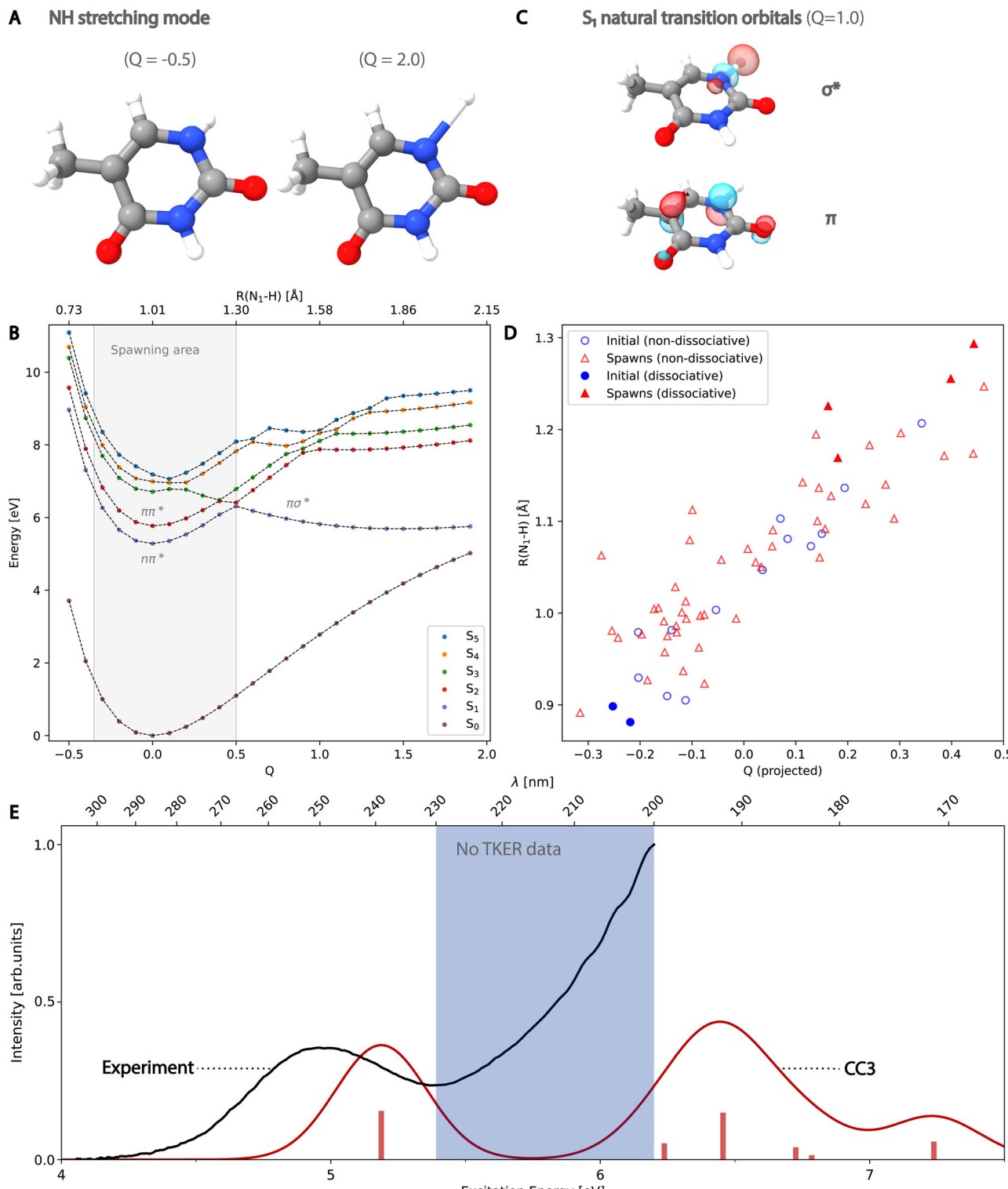

**Fig. 5 | The $\pi\sigma^*$-mediated N-H dissociation pathway.** Panel **A** illustrates the normal mode associated with the N-H stretch, where $Q$ denotes the displacement (in Bohr) along this mode relative to the ground state equilibrium geometry ($Q = 0$ corresponds to the ground state ($S_0$) minimum geometry). Panel **B** shows potential energy curves along this normal mode for the ground and first five excited states ($S_0$, $S_1$, $S_2$, $S_3$, $S_4$, and $S_5$), with the associated N-H bond length $R(N_1\text{-}H)$ given on the secondary axis. The shaded area denotes the values of $Q$ for which we see spawning events in the dynamics. Panel **C** shows the natural transition orbitals of $S_1$ for $Q = 1.0$, where the state is of $\pi\sigma^*$ character. Panel **D** shows the initial geometries (at time $t = 0$) and the spawning geometries (where population is transferred between

$S_2$ and $S_1$), distinguishing between the dissociative geometries and the non-dissociative geometries. For each geometry, we give the N-H bond length and the projected value of the mode displacement $Q$. Dissociating trajectories have negative initial values of $Q$ (short N-H distances) and spawn at high positive values of $Q$ (long N-H distances), corresponding to regions of the potential energy surfaces where the $\pi\sigma^*$ state becomes accessible. Panel **E** shows the recorded ultraviolet gas phase spectrum for thymine with a comparison to a simulated coupled cluster singles, doubles and perturbative triples (CC3) spectrum (see Suppl. Note 5). The shaded area corresponds to wavelengths where total kinetic energy release (TKER) spectra have not been recorded.

233 nm and shorter do anisotropic, fast H-atom peaks consistent with ultrafast $\pi\sigma^*$-mediated N-H dissociation appear, and this signature becomes more intense with increasing excitation energy[41]. This behavior for adenine, coupled with the presence of the $\pi\sigma^*$ N-H dissociation in our dynamics, leads us to suggest that if the TKER experiments for thymine described by Schneider and coworkers[34] are carried out using excitation wavelengths in the range 230–200 nm (as also suggested in ref. [40]), then one might observe anisotropic, fast H-atom peaks consistent with this dissociative channel. However, wavelengths in the range 230–200 nm may not be directly comparable to our simulations as it would also excite the system to states above the $S_2(\pi\pi^*)$ state, in particular, to the $\pi\pi^*$ band that lies about 1.0 eV above $S_2(\pi\pi^*)$ (see Fig. 5E and Suppl. Note 5). This is similar to adenine, where several states may contribute to the $\pi\sigma^*$ dissociation channel[41]. The utility of TKER experiments for confirming/denying the presence of the $\pi\sigma^*$ N-H dissociation channel in thymine has previously been suggested by others[40,43].

The ultimate fate of the dissociative $\pi\sigma^*$ channel cannot be resolved in this work, as CC theory is known to break down when the $\pi\sigma^*$ state becomes degenerate with the ground state (the dissociation limit). Because of this, there may also be a pathway back to the ground state through an intersection with the ground state, which might compete with N-H bond fission[40]. However, the presence and accessibility of the $\pi\sigma^*$ channel is well-described in our simulations. Both the early-time involvement of the $\pi\sigma^*$ state, and the initial rapid N-H stretch, are treated correctly with CC theory, and this aspect of the simulated dynamics paints a picture in line with both theoretical and experimental studies of $\pi\sigma^*$ states in other heteroaromatics[39,40].

## Conclusion

The photorelaxation pathways of thymine are still under debate and a clear consensus has yet to emerge, despite numerous theoretical and experimental investigations. Here, we have simulated the ultrafast dynamics of thymine using high-level nonadiabatic dynamics simulations (ab initio multiple spawning) combined with coupled cluster theory (with single and double excitations) for the electronic structure. This represents, to the best of our knowledge, the highest level simulation performed on thymine to date, and the simulation with the highest level of single-reference electronic structure on a molecular system of this size. From the data, we simulated the oxygen-edge X-ray absorption spectrum of Wolf et al.[16], obtaining excellent agreement on the appearance and position of a signal unequivocally associated with the $n\pi^*$ state, as well as for the associated $\pi\pi^*/n\pi^*$ conversion time: our theoretical estimate of $41 \pm 14$ fs agrees quantitatively with experimental estimates ($60 \pm 30$ fs and $39 \pm 1$ fs[16,17]). Furthermore, we find no significant $\pi\pi^*$ trapping.

Interestingly, our simulation predicts an additional, minor channel in which the population is rapidly transferred to a $\pi\sigma^*$ state, leading to N-H dissociation. This type of channel has been implicated in the ultrafast excited state dynamics of other heteroaromatic systems, including adenine[39,40]. In view of this surprising finding, we believe further experiments—for example, measuring the nitrogen-edge X-ray absorption spectrum—are warranted to confirm or disprove the existence of this dissociative channel as one of the relaxation pathways in thymine.

This work was made possible by recent developments in coupled cluster theory. Earlier work by some of the authors[21–23] had already indicated that the method could be modified to correctly describe conical intersections and therefore also nonadiabatic dynamics. Here, we have shown that the method introduced in these papers, the similarity constrained coupled cluster method, can in fact be applied in nonadiabatic dynamics simulations, opening up a range of applications that may now be studied with the hierarchy of coupled cluster methods.

All electronic structure methods have limitations and this is also true for coupled cluster theory. As a single-reference method, it does not give a proper description of conical intersections with the ground state, and high-order treatments are needed to describe excited states with significant double excitation character. Nevertheless, given their ability to capture dynamical correlation and to provide systematically improvable predictions, we expect that coupled cluster methods will shed new light on the photochemistry of several important systems.

## Methods
### Theoretical
The excited state dynamics simulation was performed with the ab initio multiple spawning (AIMS) method[32,33,44]. We prepared 16 initial conditions (ICs) and simulated the dynamics for a total of 4000 au (-100 fs). The initial trajectory basis functions (TBFs) were sampled from a 0 K harmonic Wigner distribution obtained from the ground state equilibrium geometry at the CCSD/aug-cc-pVDZ level. We have not applied any sampling restriction to an excitation window. Both the geometry and the frequencies were obtained with CCSD/aug-cc-pVDZ. The dynamics simulation was performed at the CCSD/cc-pVDZ level. The initial positions and momenta of the individual samples are publicly available[45]. Details about the numbering of the trajectories are given in Suppl. Note 9. For the ICs, we adopt the independent first generation approximation and average the results over 16 independent AIMS simulations. In these simulations, the $S_1$ and $S_2$ states were included in the dynamics. Of these, 12 ran normally for the whole simulation. Two ICs (the two belonging to the $\pi\sigma^*$ pathway) reach a near-degeneracy between $S_0$ and $S_1$ as the $N_1H$ bond extends beyond 2 Å and are terminated at this point because CCSD cannot describe the dissociation limit (termination is caused by convergence problems at around 20 fs). A further two ICs enter a defective region surrounding the $S_1/S_2$ intersection, at which point the energies become complex-valued and the simulations are terminated. These two ICs were re-run with similarity constrained CC theory (EOM-SCCSD) using the $\mathscr{E}$ with $T = 0$ projection[26]. For both CCSD and SCCSD, the coupling elements were evaluated with the nuclear derivative acting on the right vector without normalization; for more details, see Refs. [25,26]. In the time-integration, we have used a default timestep of 20 au (0.5 fs) and a smaller timestep of 5 au (0.1 fs) in regions of high coupling. The spawning threshold was set to 0.05, where the spawning criterion is given as the norm of the coupling times the velocity. This produced a total of 67 TBFs over the course of the simulation. Nuclear density snapshots were simulated with weights given by the squared TBF amplitudes and a 0.08 Å FWHM Gaussian broadening.

Using the data from the dynamics simulation, we simulated the oxygen-edge X-ray absorption spectrum using the CC3[42] method with the cc-pVDZ basis (additional spectra are reported in Suppl. Note 2). Core excited states were obtained with the core-valence separation approximation[46]. To simulate the spectrum, we have applied the incoherent approximation[47], where the spectrum is calculated as an average of the spectra computed at the centers of the TBFs, weighted with the corresponding TBF amplitude. To avoid state assignment ambiguities, we have used CCSD for the valence excited states and CC3 for the core excited states, where the strengths are evaluated with CCSD (excluding the approximate triples in the CC3 states). Similarly, absorption energies are evaluated from energies obtained with CCSD and CC3 (see Suppl. Note 2). All X-ray absorption spectra were shifted by $-0.5$ eV, corresponding to the required shift needed to align the first ground state peak with the experimental value at 531.4 eV. The spectra were smoothed with Gaussian broadening (10 and 100 fs FWHM and 0.3 eV FWHM) to match experimental widths and uncertainties. Some of the spectra reported in Suppl. Note 2 made use of `matplotlib`'s[48] `imshow` for Gaussian interpolations. The UV-vis absorption spectrum was calculated at the CC3/aug-cc-pVDZ+ KBJ(3–4)[49] level for the ground state equilibrium geometry,

determined with CCSD/aug-cc-pVDZ, see Suppl. Note 5. We have further simulated the nitrogen-edge X-ray absorption spectrum at Franck-Condon, at the $n\pi^*$ minimum, and at extended N-H ($Q = 1.0$), where we align the first ground state peak with experiment[50], see Suppl. Note 8. In Suppl. Notes 10 and 11, we provide spawning geometries and the stationary geometries shown in Fig. 4, respectively.

Additional simulations were performed with $S_1$ and $S_2$ included in the dynamics for 17 initial conditions with short initial NH bond lengths (see Suppl. Note 4), for the two conditions in the $\pi\sigma^*$ pathway with $S_1$, $S_2$, and $S_3$ included in the dynamics (see Suppl. Note 6), and for the initial $S_2$ dynamics of one condition in $n\pi^*$ pathway with aug-cc-pVDZ (see Suppl. Note 3).

All electronic structure calculations were performed using development versions of the $e^T$ program[51]. The AIMS dynamics was run with the FMS program and an interface to $e^T$. See Suppl. Note 12 for a description of the interface.

### Experimental

The experimental UV spectrum of thymine was taken with a Cary 5E UV-Vis-NIR spectrometer using an in-house developed gas cell described in ref. 52. The sample was purchased from Sigma-Aldrich with >99% purity and used without further refinement. The cell was heated to 150 °C to obtain sufficient absorption. Spectra were recorded over a range of 200–400 nm with a step size of 0.5 nm and an integration time of 0.5 s per data point. A background spectrum of the empty cell has been recorded at the same temperature and settings and subtracted from the spectra including the sample.

### Data availability

Data produced in this study are provided in the Zenodo repository https://zenodo.org/records/10733952. Source data for the TR-XAS and UV experimental and simulated spectra, potential energy surfaces, internal coordinates, geometries and natural transition orbitals reported in all figures and tables are provided in the Zenodo repository https://zenodo.org/records/13907876.

### Code availability

The custom code used for this study is available from the corresponding author upon request. A publicly available version of the code will be released following the time schedule of the eT program https://etprogram.org/.

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

## Acknowledgements

E.F.K., S.A., and H.K. were supported by the Norwegian Research Council through FRINATEK project 275506, the European Research Council (ERC) under the European Union's Horizon 2020 Research and Innovation Program (Grant No. 101020016). O.J.F., E.F.K., T.J.A.W., and T.J.M. were supported by the AMOS program within the U.S. Department of Energy (DOE), Office of Science, Basic Energy Sciences, Chemical Sciences, Geosciences, and Biosciences Division. OJF is a U.S. Department of Energy Computational Science Graduate Fellow (Grant No. DE-SC0023112). M.G. and D.M. were supported by DFG funding via Grant GU 1478/1-1. We acknowledge computing resources through UNINETT Sigma2–the National Infrastructure for High Performance Computing and Data Storage in Norway, project NN2962k.

## Author contributions

E.F.K., T.J.M., and H.K. conceived the project. E.F.K., S.A., and H.K. implemented the EOM-SCCSD gradients and derivative coupling elements in $e^T$. E.F.K., O.J.F., H.K., and T.J.M. implemented the FMS90-$e^T$ interface. E.F.K., S.A., and O.J.F. performed the AIMS simulations. A.C.P. simulated the time-resolved X-ray absorption spectra and UV spectra. E.F.K., S.A., O.J.F., A.C.P., T.J.M., T.J.A.W., and H.K. analyzed the simulation data. T.J.A.W. provided a new analysis of the experimental X-ray spectrum. D.M. and M.G. recorded the experimental UV spectrum. E.F.K., O.J.F., S.A., and A.C.P. wrote the first draft of the manuscript. All authors discussed the results and revised the manuscript.

## Funding

 Olavs Hospital - Trondheim University Hospital).

## Competing interests

The authors declare no competing interests.
