## [Transparent Peer Review file · Nature Communications]

Photoinduced hydrogen dissociation in thymine predicted by coupled cluster theory

Corresponding Author: Professor Henrik Koch

Version 0:

Reviewer comments:

Reviewer #1

(Remarks to the Author)

From the methodological viewpoint, this work reports a very important accomplishment. It presents the first full excited state non-adiabatic excited dynamics at the coupled-cluster level. In particular, the use of a modification of coupled-cluster theory is demonstrated, which avoids the occurrence of defective or complex-valued excitation spectra. In this respect, this article is an important demonstration of the state of the art, giving also interesting insight into the photophysical processes of thymine, in particular its fast $\pi\pi^*$ to $n\pi^*$ decay.

On the other hand, I deem it important to more clearly address the current limitations. One clear limitation is the computational cost that still imposes restrictions on the basis sets. In this work, only the minimum acceptable basis set (polarized double zeta) is employed, which is quite on the edge for excited states where additional flexibility of the basis may become important due to more likely unusual electron configurations. The resulting basis set errors clearly impede the predictive power of the method. Furthermore, the feasible level of coupled-cluster theory is still "only" CCSD. Clearly, CCSD is more robust than second-order correlation methods (which is an advantage when exploring larger parts of the potential energy surface), but it is also well-known that its accuracy is limited, as documented in a large body of literature, for a quite comprehensive one, see, e.g. DOI 10.1021/acs.jpca.1c08524 and related work. Lastly, one also should be clear that CCSD and other single-reference coupled-cluster methods rely on a solid description of the ground state reference by a single determinant. In that respect, I disagree with the statement that the method used in this work is generally applicable to "photochemical phenomena" (line 86). This is raising too high expectations, as CCSD clearly cannot dissociate bonds (which in photochemistry likely happens). The same applies for the final statement in line 285. In my view the application area has to be restricted to photophysical effects, which mainly covers non-radiative decays (which still leaves a broad range of applications). All these aspects should be disclosed in a way that gives a clear guideline to the general audience with more limited expertise in quantum chemistry.

With respect to the accuracy and robustness of the predictions made in this work, the above paragraph makes (hopefully) clear, why I am a bit reluctant. To make a more convincing prediction, I would expect some studies that explore the variation of predictions by increasing the accuracies of the methods for certain important pathways along the PES (by, e.g., larger basis set, including triply excited clusters, counterchecking with multireference methods) and giving estimates, how strongly the observed uncertainties influence the predicted outcomes. I am also wondering, if the rating "unexpected" in the title is justified, as the presence and involvement of $\pi\sigma^*$ states in nucleobases and related compounds has been discussed for more than a decade, as it is also clearly admitted in the text. In that respect, this is not a novel and stunning pathway, the question is rather how accurate the predictions of the current work are and whether this improves on previous works on the presence or absence of this path.

Taking all this together, I think that this is a very nice and well-written work demonstrating a clear step forward in the state of the art and I think that this deserves highlighting in a top-tier journal. But I think that my concerns in the previous two paragraphs need consideration.

In addition, let me here give a list of further comments:

The statements about the first application of the method in nonadiabatic dynamics are duplicated in lines 76/77 and 83-85.

Line 110: "recent experimental and theoretical investigations have not invoked this pathway ...": references are missing

Fig 3: Understanding panel B and D correctly requires some technical background. I understand that wavepackets are propagated, but still a definite potential energy is assigned to each time step; is this the energy at the center of the wavepacket? And the two curves are given for the same nuclear configuration? Maybe a quick clarification (or reference to supporting information with more explanation there) could be helpful.

I was also confused by the discussion in line 158 and beyond. Why is it said that the wavepacket assumes $\pi\pi^*$ or $n\pi^*$ character (which in the first place are descriptors of the electronic state). I am also wondering, if this discussion is rather technical or if it really is of fundamental importance here?

What exactly are "CCSD-to-CC3 energy differences" (line 318)? I assume this cannot be plain energy differences of CCSD and CC3 total energies, as this would involve a biased description of correlation energies.

Line 296 mentions two trajectories that end up in a S1/S0 intersection (and, as reported, break off due to convergence issues): Wouldn't that be an as important pathway as the highlighted $\pi\sigma^*$?

Reviewer #2

(Remarks to the Author)

In the manuscript "Unexpected hydrogen dissociation in thymine: predictions from a novel coupled cluster theory", the authors Kjørnstad et al. present the first application of a novel excited-state electronic structure method to nonadiabatic dynamics simulations. They apply the method to thymine in the gas phase and find a yet unknown hydrogen dissociation channel. They also provide an up-to-date theoretical description of the dynamics of the pipi^* and npi^* states, which have been intensively debated in the last decades. Given all of that, the topic and findings of the manuscript appear to be highly relevant and would clearly have a significant impact on the community.

Unfortunately, the manuscript currently has several substantial issues. First, the writing quality of the manuscript is very inconsistent. The introduction already prematurely reveals all core results but without context. The results section requires much better guiding of the readers. The supporting information and its referencing in the main text are, unfortunately, very chaotic. Second, I have strong doubts about the correctness of the simulations regarding the N-H dissociation channel, as detailed below. I also have several minor issues that the authors hopefully can fix.

Overall, I think that the manuscript - if the NH dissociation results are correct - is certainly of high interest and significance and it is great to see a new excited-state nonadiabatic method in action. Also the quality of the simulated spectra seem to be of very high quality. Nevertheless, while I would like to see the manuscript published, this is not possible in the current form.

Major issues

1) The most severe problem of the manuscript is that the hydrogen dissociation process, which is already emphasized in the title, might be simulated in an inaccurate way. I first became suspicious about this in the description at the top of page 9 (related to Figure S14). The authors write that at 2.4fs, the S2 is the pipi^* state but at 4.0fs the S2 is the pisigma^* state. At 4.1fs, the population is transferred to the S1 state.

Assuming that the wavepacket starts in the S2 state (never explicitly stated), it appears that over the course of less than 2fs, the electronic wavefunction evolves from pipi^* to pisigma^* , which would be extremely fast. My suspicion is rather that the AIMS simulations only included S1 and S2, but not S3, as all population plots show only these two states (number of states is never given). If the S3 was included in the simulation, then at the S2/S3 conical intersection, a large fraction of the wavepacket should have transferred to the S3 in order to keep the pipi^* character. But if S3 is neglected, then all S2->S3 population transfers are automatically suppressed and the S2 population is forced to change character from pipi^* to pisigma^* in an unphysical way.

The authors need to provide simulations of the dissociating initial conditions that include the S3 state. Only if the simulations including S3 also show strong population transfer to the pisigma^* , then the hydrogen dissociation mechanism can be trusted.

2) There are multiple problems with the writing of the manuscript that could lead to confusion of the readers.

From the flow of the text, one obtains the impression that the Results and Discussion section should actually start at the top of page 4. There, the first new results are already presented, in particular Figure 2, which shows the simulated O-edge XANES results that fit very well with experiment. Then, the authors tease the existence of the hydrogen dissociation channel at the end of the introduction. In the Results and Discussion, the authors start again at Figures 1 and 2, repeating information that was already written in the figure captions.

I strongly suggest that the authors clean up the introduction to avoid repetition and to help readers separate previous work from the current work. Instead, the authors could use the space in the introduction to provide some essential information about the simulations, see my point 10) below. It might also be useful to formulate the goals of the study in the introduction. The rest of the Results and Discussion would also benefit strongly from better guides to the readers. Subheadings should be

added to separate the discussion of electronic/spectroscopic results, nuclear motion results, and the new hydrogen dissociation channel.

3) Closely connected with point 2), the authors need to make an effort to better organize and discuss their supplementary results. The supplementary information has 9 sections, 18 figures, and 4 tables. The only items that are mentioned in the main text are Sections S2, S1, S6, S7, and S9, in that order. Not a single supplementary figure or table is referenced. Even the few existing references are not very helpful, because within the supplementary information, referencing is also very bad. Section S1 mentions Figures S1-S5, and then the next reference is Figure S15 in Section S3. Figures S6-S12 are never mentioned (one can guess that they belong to Section S2). Figures S13 and S14 seem to belong to Section S6 (along with Figure S16). Figure S18 and Table S4 are never mentioned either (do they belong to Section S9?).

Furthermore, the sections and figures/tables are badly aligned, for example does Figure S8 - belonging to Section S2 - only appear after Section S9.

Moreover, there are 4 citations that only appear in the supplementary information, so they will not be indexed and their authors will not receive the appropriate credit. One of the citations is even to the authors' own Zenodo repository with additional simulation data, so the authors should have an interest that readers actually find this citation.

I strongly recommend that the authors order the supplementary information correctly (please look up the latex command `\clearpage`). All supplementary items (sections, figures, tables) should be referenced in the main text in the correct order and within the supplementary information. I also suggest to add a table of content to the SI and to put all citations in the main text.

4) Figure 1 is a very large figure that provides only very limited insight and appears almost decorative. One can observe that there are two potential energy surfaces, but the interesting crossing region is hardly visible due to the chosen colors and transparency. It is unclear whether this is a sketch or a real PES scan, and what the degrees of freedom are. It is also somewhat deceptive, because later the authors mention three essential degrees of freedom (r_{48} , r_{56} , and r_{NH}), but Figure 1 gives the impression that there are only two. The orbital plots, which are in principle useful to introduce the electronic states to the readers, are rather small instead.

As mentioned above, a further problem is that the caption already preempts several results from below, increasing the confusion whether the figure is a sketch or not.

5) Figure 3 appears somewhat out of place in the main manuscript. In contrast to the other main figures, Figure 3 does not provide any insight into the photodynamics of thymine. Given how the title, abstract, and introduction lead the readers into the paper, the figure therefore does not fit into the narrative. Possibly even the authors have realized this, because the Figure is only mentioned very briefly once: "Close to the intersection, CCSD exhibits numerical artifacts which can be removed with SCCSD. Figure 3 shows one of these artifacts as well as how it is corrected."

While I can understand the authors that they want to present the strengths of the new method, I do not have the impression that this figure is the best way to do so. It also does not help that the SCCSD method was actually only used for 2/16 simulations, whereas CCSD was used for the remainder. They should either expand the discussion of the figure (e.g., in a separate subheading at the end of the Results and Discussion) or move the figure to the SI.

6) One of the striking results of the new study is the simulated time-resolved O-edge XANES in Figure 2. It lends significant credibility to the results of the study (albeit not about the NH dissociation). Because of this, I find it a pity that the authors did not apply the same level of attention to the N-edge XANES, instead hiding it in the supporting information. If the NH dissociation channel is real and the authors want to stimulate further experiments on thymine, then it would only be consequential to simulate the time-resolved N-edge XANES and show it in the main text.

Minor issues

7) Abstract and introduction: The authors currently do not mention in any place whether they talk about the dynamics of thymine in gas-phase, in solution, or in some biological environment. Only readers that are aware of the experimental methods or that know the citations can conclude that the manuscript is about gas-phase dynamics. This information is critical, because thymine dynamics is significantly different in different environments. Especially N-H dissociation will be very different in condensed phase. Please add appropriate hints in the abstract and introduction (and possibly the title).

8) Introduction: "Different theoretical methods have therefore produced different explanations, and a consensus has yet to emerge." Are the authors here referring to the next paragraph? They might want to add the citations already to this sentence, to make sure what they are referring to. Furthermore, a statement could be added that most of the non-consensus is due to different electronic structure levels of theory.

9) While paragraphs 2 and 3 nicely summarize a lot of experimental and theoretical evidence on thymine, they completely spare out intersystem crossing. This probably does not meaningfully affect the simulations, but for completeness the few-% triplet yields in thymine should be mentioned.

10) Although it is fine that the bulk of the computational details is collected in the Methods section at the end, the most important pieces of information should also be given in the main text, else the reader will constantly need to leave back and forth. Some important details that should be added to the beginning of the Results and Discussion (or introduction) are the following.

(i) The simulations are actually performed with EOM-CCSD/EOM-SCCSD. Most readers will probably assume ground state calculations if only "CCSD" is written, or will wonder what excited-state theory is used (response theory like in CC2, EOM, delta-CC, etc).

(ii) That the nonadiabatic dynamics theory is multiple spawning.
(iii) That the simulation time was 100fs. In this context, one should also mention that even SCCSD cannot describe S1/S0 conical intersections, so longer simulation times would not have been useful.

11) The percentages in the caption of Figure 1 indicate that only two pathways are present, trapping in $\text{np}\pi^*$ (87%) and NH dissociation (13%). This finding requires some contextualization, as many previous studies have found/indicated that a large fraction of population goes directly from the $\text{p}\pi\pi^*$ to the ground state, with a short stretch where the $\text{p}\pi\pi^*$ is S1.

12) "Moreover, we find frequency oscillations" What does this mean? Should it simply read "we find oscillations" or rather "we find oscillations in the frequency of some signal"?

13) The discussion in the top paragraph of page 7 about the difference between adiabatic and diabatic populations (and related time constants) is not really new anymore. There are many works in the literature that show that adiabatic populations are not useful to compare to experimental observables. Maybe the discussion could be rephrased such that readers do not get the impression that this aspect is new. Additionally, it might be useful to show the electronic populations (any) in the main manuscript, maybe in Figure 2.

14) In the caption of Figure 3: "Dynamics with CCSD (panels A and B) and SCCSD (panels C and D)." I think this caption is not appropriate. Panels A and C do not show dynamics, but a plot of a conical intersection. The fragment "branching or gh plane for an initial condition" is even more confusing, because an initial condition does not have a branching plane.

15) In Figure 4, it is somewhat hard to see how the wavepacket is actually moving. I recommend to add some arrows to guide the eye. Furthermore, please indicate whether the total wavepacket is shown or only the one on the S2 surface (or S1?).

I am also not really sure what the message of this figure should be. The main text discussed the figure only briefly in a short paragraph on page 7. Maybe this could be better worked out, including a comparison to other literature works?

16) In Figure 5, I was wondering why the authors employ a very high level of theory for the absorption spectrum (CC3 with a basis set tailored for Rydberg states) but then simply obtain the spectrum by Gaussian convolution, where they did not even match the Gaussian width to the first absorption band. Why was the spectrum not computed for the Wigner distribution of initial conditions? In this way, agreement with experiment could be strongly improved.

17) Conclusion: In my opinion, "with the highest level of electronic structure" is a controversial statement here. Many researchers will disagree that a method that cannot describe S1->S0 dynamics is "the highest level" for a system like thymine, where this dynamics is very important. My point here is that excited-state electronic structure methods are too diverse that one can simplify them into a one-dimensional scheme where one is clearly the "highest". Hence, I suggest to rephrase.

Also note that other high-level nonadiabatic dynamics for nucleobases have been reported before, e.g., DOIs 10.1021/acs.jpcclett.1c00926, 10.1111/php.13922, or 10.1021/acs.jpcclett.1c00712.

Reviewer #3

(Remarks to the Author)
please see attached file

Reviewer #4

(Remarks to the Author)

The manuscript the first ab initio on-the-fly simulation of dynamics through conical intersections utilizing the coupled-cluster singles and doubles (CCSD) method. The article addresses and clarifies a long-standing controversy surrounding the dynamics following excitation of the lowest $1\text{p}\pi\pi^*$ state of thymine. Furthermore, the study confirms, for the first time via computation, the existence of a competing relaxation channel via the lowest $1\text{p}\sigma\pi^*$ state for a pyrimidine base. This significant discovery not only advances our understanding of pyrimidine base behavior but also broadens our comprehension of electronic structure and dynamics in molecular systems.

In summary, the manuscript represents a methodologically robust and intellectually stimulating contribution to the field of computational chemistry. The significance of its findings, combined with the rigorous computational approach employed, makes it an excellent fit for publication in Nature Communications. Therefore, I strongly recommend publication of the paper if the following points are clarified by the authors.

Technical comments:

(1) Fig. 1 is not correct in this form. Either the potential energy surfaces (PES) of a three-state intersection (S1, S2, S3) should be shown, or the $\text{p}\sigma\pi^*$ channel should be removed. The intersections between three states are clearly seen in Fig. 5E.

(2) The discussion of the nonadiabatic dynamics, especially on p. 7, is confusing. The notation S2($\text{p}\pi\pi^*$), S1($\text{np}\pi^*$) and S3($\text{p}\sigma\pi^*$) does not make sense when there are multiple intersections of adiabatic PES in the vicinity of the Franck-Condon (FC) zone, e.g. in Fig. 1. Adiabatic surfaces are defined as S1, S2, S3 and are ordered by energy everywhere.

Diabatic surfaces are defined as $1\pi\pi^*$, $1\pi\pi^*$, $1\pi\sigma^*$ and are not ordered by energy. Adiabatic surface S_n can have $1\pi\pi^*$, $1\pi\pi^*$ or $1\pi\sigma^*$ electronic character, depending on the nuclear geometry.

(3) The formulation on p. 7: "the adiabatic populations overestimate the true rate of internal conversion" turns the logic on the head. The molecular observables are the electronic populations and the time derivatives of the adiabatic populations are the rates. Depending on the situation (PES, transition dipole moments, type of signal), the time-resolved signal may approximately report adiabatic populations, diabatic populations or none of them. Adiabatic and diabatic populations are related by a geometry-dependent unitary transformation, as has been discussed since decades in the quantum wavepacket literature and more recently also in on-the-fly surface-hopping dynamics, e.g. Zhou et al., JPCLet. 2019, 10, 7062 or Xie et al., JCP 150, 154119 (2019). The discussion in Section S1 of the SI should be embedded in the existing literature.

(4) The TRPE experiment of Suzuki and coworkers (Miura et al., Ref. 35) stands out by its excellent time resolution and excellent signal-to-noise ratio. It first definitively established the lifetime of the $\pi\pi^*$ state of thymine. This work should be discussed in the introduction rather than on p. 9.

(5) The term "wavepacket dynamics" is used in the abstract and throughout the manuscript. It should succinctly be explained whether 16 classical trajectories (of which 10 survive) with overall 67 Gaussian basis functions do accurately represent quantum wavepackets which are split at multiple conical intersections.

(6) Caption Fig. 2: The energy and time resolutions of transient absorption (TA) pump-probe (PP) spectra are Fourier limited. In gas-phase samples (no inhomogeneous broadening) the energy resolution is completely determined by the pulse durations. What is the justification for the Gaussian broadening in time and the additional Gaussian broadening of 0.3 eV in energy? These important details of the simulations should be explained in the Theoretical Methods section or in a section of the SI rather than in the figure caption.

(7) The non-augmented cc-pVTZ basis is inappropriate for the $\pi\sigma^*$ PES in the vicinity of the FC zone, where the σ^* orbital is very diffuse. The parts of the adiabatic PES which are of $\pi\sigma^*$ character likely are too high in energy. This limitation of the simulations should be discussed.

(8) Having learned that the CCSD method describes the PES of thymine well, the reader also would like to know what went wrong in the previous simulations, e.g. Ref. 6. Is it the lack of dynamical correlation energy in the CASSCF method?

Version 1:

Reviewer comments:

Reviewer #1

(Remarks to the Author)

The authors have submitted a thoroughly revised version of their manuscript and the criticism of the reviewers has been discussed in great detail in their rebuttal (and is echoed in their revised manuscript).

I have only a few minor comments, otherwise I can clearly recommend this work for publication.

Concerning the last comment (#8) of reviewer 4, I feel that this is a legitimate (and in way also central) question, which boils down to "can we understand why the results of the present study are different (and better) than previous work" (where the pure "coincidence with experiment" is maybe not the most convincing argument)? As there is an overlap of the present author team to the authors of one of the earlier studies, which employs rather orthogonal techniques like CASSCF (and MS-CASPT2), I could expect that a more detailed answer could be given. It would be (as I think) also worthwhile to point out that findings like the $\pi\sigma^*$ channel are more difficult in a CASSCF-based approach, as the active space choice may exclude certain channels right from the outset.

Just as a comment that does not require further action by the authors: Fig. S20 shows that the $\pi\sigma^*$ pathway is a stable feature (and that is what the authors also state in the main text), but it also clearly shows that the basis set limit is far away and that the prediction of excited state dynamics (in particularly at higher energies with increasing density of states) remains an adventure.

Fig. 3: In A and C, the units of g and h are not defined (I assume unitless normal coordinates, but it would be better, if this is stated). For panels B and D, I suggest to convert the abscissa to eV (to match with other graphs).

Fig. 5: In principle, I am OK with the shaded area, but there is one thing that creates confusion. The experimental spectrum is labelled with "Experiment" and a dotted line connects the label with the graph of the experimental spectrum, coming from outside the shaded area and ending in this area. This (at least for me) gives at first glance the impression that the label "Experiment" refers to the shaded area. I suggest to move this label to a different position and to also explicitly label the shaded area with, e.g., "no TKER data" or something similar.

Again Fig. 5 (and text on page 7): I assume that Q is expressed in unitless (=mass and frequency weighted) coordinates. This might be added at an appropriate place.

I also noted that the authors partially use "au" as "atomic units" (mainly in the technical part and in the SI), but at the same time in Fig. 5 "a.u." stands for arbitrary units (absorption). I am normally a big fan of spelling out atomic units, so the atomic unit of time, for instance, is written as \hbar/E_{H} . This may also be helpful to reader to understand the translation to femtoseconds.

I also noted that in some instances in the SI, the atomic unit of time is used, where the reference to the content of the graphic is only indirect, as there fs are used as units, see Figs. S14 and S15.

In Section S9 "au" is used for lengths (which is confusing, as "tau" used as the symbol is often associated with time), I suggest to use a_0 here (the official symbol for Bohr units).

Reviewer #2

(Remarks to the Author)

In their revised manuscript, the authors have greatly improved the manuscript and solved the issues raised by the reviewers. The manuscript can now in principle be published in Nature Communications. I have only a few smaller comments, as given below:

- 1) Despite it being many items, I would recommend the authors to reference all SI items in the main text, and not only the sections. I believe that this is actually Nature Communications policy, so the authors might need to add such references at some point anyways.
- 2) Based on the replies in the rebuttal, I have the impression that the authors consider employing the new electronic structure method as one of the main messages of the manuscript. In that case, I strongly suggest to include this message in the abstract, where it is currently missing.
- 3) I suggest to add to the statement of "We find no significant $\text{p}i\text{p}i^*$ trapping and no direct $\text{p}i\text{p}i^*$ relaxation to the ground state." on page 4 the time scale of 100fs, to avoid any possible misunderstandings in the readers. A reminder on the limitations of the level of theory is found elsewhere, but could be added in this sentence as well.
- 4) I still do not fully understand with the several general statements claiming the "highest electronic structure level of theory". As I stated before, I believe that one cannot simply line up all levels of theory and define a clear "highest". At the very least, the authors should modify the claim to "highest single-reference level of theory", which is a statement with which I could agree.

Reviewer #4

(Remarks to the Author)

I'm satisfied by authors response to my comments and corrections in the MS.

Response to reviewer's comments for
“Unexpected hydrogen dissociation in thymine:
predictions from a novel coupled cluster theory”

Eirik F. Kjøenstad^{*1,2,3}, O. Jonathan Fajen^{1,2}, Alexander C. Paul³,
Sara Angelico³, Dennis Mayer⁴, Markus Gühr^{4,5},
Thomas J. A. Wolf¹, Todd J. Martínez^{*1,2}, Henrik Koch^{*3}

¹Department of Chemistry, Stanford University, Stanford, CA, USA.

²Stanford PULSE Institute, SLAC National Accelerator Laboratory,
Menlo Park, CA, USA.

³Department of Chemistry, Norwegian University of Science and
Technology, Trondheim, 7491, Norway.

⁴Deutsches Elektronen-Synchrotron DESY, Hamburg, Germany.

⁵Institute of Physical Chemistry, University of Hamburg, Hamburg,
Germany.

Contributing authors: eirik.kjonstad@ntnu.no;
todd.martinez@stanford.edu; henrik.koch@ntnu.no;

We thank all the reviewers for their comments, which we address point-by-point below and which we believe have led to significant improvements of the manuscript.

Reviewer 1

Comment: From the methodological viewpoint, this work reports a very important accomplishment. It presents the first full excited state non-adiabatic excited dynamics at the coupled-cluster level. In particular, the use of a modification of coupled-cluster theory is demonstrated, which avoids the occurrence of defective or complex-valued excitation spectra. In this respect, this article is an important demonstration of the

state of the art, giving also interesting insight into the photophysical processes of thymine, in particular its fast $\pi\pi^*$ to $n\pi^*$ decay.

Response: We thank the reviewer for their evaluation that the work represents an accomplishment in method development while also providing interesting insights into the photophysics of thymine.

Comment: On the other hand, I deem it important to more clearly address the current limitations. One clear limitation is the computational cost that still imposes restrictions on the basis sets. In this work, only the minimum acceptable basis set (polarized double zeta) is employed, which is quite on the edge for excited states where additional flexibility of the basis may become important due to more likely unusual electron configurations. The resulting basis set errors clearly impede the predictive power of the method. Furthermore, the feasible level of coupled-cluster theory is still "only" CCSD. Clearly, CCSD is more robust than second-order correlation methods (which is an advantage when exploring larger parts of the potential energy surface), but it is also well-known that its accuracy is limited, as documented in a large body of literature, for a quite comprehensive one, see, e.g. DOI 10.1021/acs.jpca.1c08524 and related work. Lastly, one also should be clear that CCSD and other single-reference coupled-cluster methods rely on a solid description of the ground state reference by a single determinant. In that respect, I disagree with the statement that the method used in this work is generally applicable to "photochemical phenomena" (line 86). This is raising too high expectations, as CCSD clearly cannot dissociate bonds (which in photochemistry likely happens). The same applies for the final statement in line 285. In my view the application area has to be restricted to photophysical effects, which mainly covers non-radiative decays (which still leaves a broad range of applications). All these aspects should be disclosed in a way that gives a clear guideline to the general audience with more limited expertise in quantum chemistry.

Response: We thank the reviewer for pointing out limitations imposed by both the correlation treatment and the basis set description, and we agree that such effects can be important for an accurate description of the excited state potential energy surfaces. We have adjusted the conclusion to make limitations clear to a general audience:

All electronic structure methods have limitations and this is also true for coupled cluster theory. As a single-reference method, it does not give a proper description of conical intersections with the ground state, and high-order treatments are needed to describe excited states with significant double excitation character. Nevertheless, given their ability to capture dynamical correlation and to provide systematically improvable predictions, we expect that coupled cluster methods will shed new light on the photochemistry of several important systems.

We do not include a discussion of the basis set here, as this is not directly related to the methodological advances presented. The effects of the basis set in our simulation are addressed below.

We agree that the method is not applicable to all "photochemical phenomena" and it was not our intention to imply this. We have adjusted the sentence to clarify:

This demonstrates that coupled cluster theory is a viable electronic structure method for simulating a range of photochemical processes.

We believe a restriction to “photophysical” phenomena is too limited, however. The reviewer is correct to point out that bond-breaking processes are difficult to describe, but these processes become feasible in principle (though at increased cost) with higher-order coupled cluster methods. Moreover, a less accurate description in the dissociation limit does not preclude an accurate description before this limit is reached.

Comment: With respect to the accuracy and robustness of the predictions made in this work, the above paragraph makes (hopefully) clear, why I am a bit reluctant. To make a more convincing prediction, I would expect some studies that explore the variation of predictions by increasing the accuracies of the methods for certain important pathways along the PES (by, e.g., larger basis set, including triply excited clusters, counterchecking with multireference methods) and giving estimates, how strongly the observed uncertainties influence the predicted outcomes. I am also wondering, if the rating “unexpected” in the title is justified, as the presence and involvement of $\pi\sigma^*$ states in nucleobases and related compounds has been discussed for more than a decade, as it is also clearly admitted in the text. In that respect, this is not a novel and stunning pathway, the question is rather how accurate the predictions of the current work are and whether this improves on previous works on the presence or absence of this path.

Taking all this together, I think that this is a very nice and well-written work demonstrating a clear step forward in the state of the art and I think that this deserves highlighting in a top-tier journal. But I think that my concerns in the previous two paragraphs need consideration.

Response: We thank the reviewer for the positive assessment of our work and for recommending that it should be highlighted in a top-tier journal.

The label “unexpected” is in our view appropriate for two reasons. The first is that the pathway has not been identified in other dynamics simulations on thymine, and that the discussion has, as a result, centered on the $\pi\pi^*$ and $n\pi^*$ states. The second is that there is experimental evidence [1] to the contrary.

The revised manuscript includes additional calculations that indicate that the predicted pathways are robust. First, we have explicitly included the S_3 state in the two initial conditions that follow the dissociative $\pi\sigma^*$ channel. The simulation results are unchanged, showing that including the S_3 state does not suppress the $\pi\sigma^*$ pathway. The revised text reads:

[...] Both of the dissociative initial conditions display identical behavior when the S_3 state is included in the dynamics simulation, showing that its inclusion does not suppress the channel (see Supporting Information S6) [...]

Second, we have calculated potential energy curves along the N_1 -H coordinate using perturbative triple excitations (CC3/cc-pVDZ) and with a basis set with diffuse functions (CCSD/aug-cc-pVDZ). In the revised text, we write:

[...] Moreover, while the $\pi\sigma^$ state has Rydberg character in the Franck-Condon region, this character decreases as the N_1 -H bond extends and the state becomes involved in the*

dynamics. Potential energy curves along the N_1 -H bond with a diffuse basis set (aug-cc-pVDZ) and with a higher-order correlation treatment (coupled cluster with perturbative triples, CC3 [2]) suggest that the pathway is still present with more accurate treatments (see Supporting Information S7).

Third, we have performed early-time dynamics for one initial condition (IC 5) using aug-cc-pVDZ. This particular IC is part of the $\pi\pi^*$ to $n\pi^*$ channel. We find that the dynamics from the Franck-Condon region to the S_1/S_2 intersection is very similar. In particular, both simulations (using cc-pVDZ and aug-cc-pVDZ) provide similar time evolution of the C_5 - C_6 and C_4 - O_8 bonds and reach the spawning region at around the same time (≈ 450 au). The revised text reads:

[...] In Figure 4, we show the time evolution of the nuclear density in the C_5 - C_6 and C_4 - O_8 bond coordinates, and we indeed see this two-step process unfolding in real time. For an initial condition simulated using an extended basis set (aug-cc-pVDZ), we also find that the initial dynamics follows the same path from the Franck-Condon region to the S_1/S_2 intersection region (see Supporting Information S3).

Comment: In addition, let me here give a list of further comments:

The statements about the first application of the method in nonadiabatic dynamics are duplicated in lines 76/77 and 83-85.

Response: We thank the referee for pointing out this repetition. The paragraph has been revised:

Here we present the highest-level wavepacket simulation on the early dynamics of gas phase thymine to date. To the best of our knowledge, this is also the simulation with the highest level of electronic structure theory performed on a molecular system of this size. Thanks to recent developments [3–8], we were able to describe the electronic structure with the highly accurate coupled cluster singles and doubles (CCSD) [9] method in its equation of motion formulation for excited states (EOM-CCSD) [10]. This is the first time that this method is applied in nonadiabatic dynamics. Coupled cluster (CC) theory is well-known for effectively capturing dynamical correlation, but it has been widely regarded as unsuited for excited state dynamics due to the presence of numerical artifacts at conical intersections [3, 11–13]. Recent work has shown that these problems can be removed with similarity constrained CC (SCC) theory [4, 5, 8]. Here, we apply the SCC with singles and doubles (EOM-SCCSD) method to simulate the first 100 fs after photoexcitation using ab initio multiple spawning (AIMS) [14, 15], showing that it is possible to simulate nonadiabatic dynamics with a coupled cluster method that correctly describes conical intersections. This demonstrates that coupled cluster theory is a viable electronic structure method for simulating a range of photochemical processes.

Comment: Line 110: "recent experimental and theoretical investigations have not invoked this pathway ...": references are missing

Response: We thank the referee for pointing this out. We have added references:

[...] showed no signature of ultrafast N-H dissociation [1], and recent experimental and theoretical investigations (see, e.g., Refs. 16–19) have not invoked this pathway to explain the molecular dynamics [...]

Comment: Fig 3: Understanding panel B and D correctly requires some technical background. I understand that wavepackets are propagated, but still a definite potential energy is assigned to each time step; is this the energy at the center of the wavepacket? And the two curves are given for the same nuclear configuration? Maybe a quick clarification (or reference to supporting information with more explanation there) could be helpful.

Response: We agree that more detail should be added here and we have adjusted the caption in the revised text. In the AIMS method, the nuclear wavepacket is expanded in terms of nuclear trajectory basis functions, see, e.g., Refs. 14, 15. In Panels B and D, we show the potential energy curves for the center of one of these basis functions. The revised caption reads:

An example of a numerical artifact encountered during the dynamics with EOM-CCSD (panels A and B) and their correction with EOM-SCCSD (panels C and D). Panels A and C show a branching or gh plane for a conical intersection encountered during the simulation of one of the initial conditions. ΔE is the energy difference between S_1 and S_2 . For CCSD (A), there is an ellipse of degeneracy in the gh plane with unphysical complex-valued energies in the interior of the elliptical boundary; for SCCSD (C), we instead see a single point of degeneracy and no unphysical energies. In CCSD simulations, the wavepacket may approach the intersection too closely and end up in the region with complex-valued energies (the interior of the ellipse in the gh plane). Whenever this happens, we re-run the simulation with SCCSD. Panels B and D show the corresponding potential energy curves for the center of a nuclear trajectory basis function with CCSD (B) and the same trajectory basis function with SCCSD (D). At 71 fs, the CCSD simulation enters the complex-valued region and is terminated (B). The SCCSD simulation, on the other hand, does not encounter any problems (D). The expansion point used in the branching plane calculation is the geometry with the smallest ΔE (as given by SCCSD) in the nonadiabatic event at 71 fs.

Comment: I was also confused by the discussion in line 158 and beyond. Why is it said that the wavepacket assumes $\pi\pi^*$ or $n\pi^*$ character (which in the first place are descriptors of the electronic state). I am also wondering, if this discussion is rather technical or if it really is of fundamental importance here?

Response: We thank the reviewer for bringing this potential source of confusion to our attention. When we referred to the character of the wavepacket, we were referring to the effective character obtained by averaging over the nuclear density. More precisely, this is the effective electronic character of the molecular wave function. We have adjusted the language throughout. The revised paragraph now reads:

The $\pi\pi^/n\pi^*$ conversion time of $\tau = 41 \pm 14$ fs was determined by analyzing the growth of the 526 eV signal in the simulated spectrum. This time constant is consistent with the rate of $\pi\pi^*/n\pi^*$ conversion in the simulated dynamics, that is, from the observed change in electronic character from $\pi\pi^*$ to $n\pi^*$. We find a rapid adiabatic population transfer from S_2 to S_1 ($\tau = 17 \pm 1$ fs) in our simulation. However, when the adiabatic states are decomposed into their diabatic components, and in particular into their $\pi\pi^*$ and $n\pi^*$ components, we see that the growth in the $n\pi^*$ character ($\tau = 37 \pm 9$ fs) is in close agreement with the*

time constant determined from the simulated spectrum ($\tau = 41 \pm 14$ fs). This shows that the 526 eV signal in the spectrum is due to the electronic $n\pi^*$ character.

Comment: What exactly are “CCSD-to-CC3 energy differences” (line 318)? I assume this cannot be plain energy differences of CCSD and CC3 total energies, as this would involve a biased description of correlation energies.

Response: We agree with the reviewer that this point requires further elaboration, and we now provide additional details in Supporting Information S2:

[...] Using CCSD for the valence states ensures that the valence excited states in the dynamics simulation match the valence excited states in the X-ray absorption calculations. From the CCSD and CC3 calculations, we define the energies

$$E(S_0) = E_{\text{CCSD}}(S_0) \quad (1)$$

$$E(S_n^{\text{val}}) = E_{\text{CCSD}}(S_0) + \omega_{\text{CCSD}}(S_n^{\text{val}}) \quad (2)$$

$$E(S_n^{\text{core}}) = E_{\text{CC3}}(S_0) + \omega_{\text{CC3}}(S_n^{\text{core}}) \quad (3)$$

and the energy differences

$$\begin{aligned} E(S_n^{\text{core}}) - E(S_n^{\text{val}}) &= \omega_{\text{CC3}}(S_n^{\text{core}}) - \omega_{\text{CCSD}}(S_n^{\text{val}}) + [E_{\text{CC3}}(S_0) - E_{\text{CCSD}}(S_0)] \\ E(S_n^{\text{core}}) - E(S_0) &= \omega_{\text{CC3}}(S_n^{\text{core}}) + [E_{\text{CC3}}(S_0) - E_{\text{CCSD}}(S_0)] \end{aligned} \quad (4)$$

This produces an effective shift in both types of energy differences (valence-to-core, ground-state-to-core) equal to $E_{\text{CC3}}(S_0) - E_{\text{CCSD}}(S_0)$. Since we shift the spectra to align the ground state peak, and since the variation of $E_{\text{CC3}}(S_0) - E_{\text{CCSD}}(S_0)$ is small, this does not have a significant effect on the simulated spectra. All spectra were shifted by -0.5 eV to match the experimental ground state bleach at 531.4 eV [20].

Note also that the peak positions are similar to those obtained with a valence and core CC3 simulation of the spectrum. However, combining CCSD with CC3 (as described above) has the benefit of allowing us to avoid the state assignment problem (i.e., we avoid having to decide which CC3 valence state corresponds to which CCSD valence state, which is not always possible and leads to inconsistencies).

Comment: Line 296 mentions two trajectories that end up in a S1/S0 intersection (and, as reported, break off due to convergence issues): Wouldn't that be an as important pathway as the highlighted $\pi\sigma^*$?

Response: The two initial conditions mentioned on line 296 are the ones that follow the $\pi\sigma^*$ pathway. The revised text reads:

Two ICs (the two belonging to the $\pi\sigma^*$ pathway) reach a near-degeneracy between S_0 and S_1 as the N_1H bond extends beyond 2 Å and are terminated at this point because CCSD cannot describe the dissociation limit (termination is caused by convergence problems at around 20 fs).

Reviewer 2

Comment: In the manuscript “Unexpected hydrogen dissociation in thymine: predictions from a novel coupled cluster theory”, the authors Kjøenstad et al. present the first application of a novel excited-state electronic structure method to nonadiabatic dynamics simulations. They apply the method to thymine in the gas phase and find a yet unknown hydrogen dissociation channel. They also provide an up-to-date theoretical description of the dynamics of the pipi^* and npi^* states, which have been intensively debated in the last decades. Given all of that, the topic and findings of the manuscript appear to be highly relevant and would clearly have a significant impact on the community.

Response: We thank the reviewer for the positive evaluation of the work’s significance and relevance.

Comment: Unfortunately, the manuscript currently has several substantial issues. First, the writing quality of the manuscript is very inconsistent. The introduction already prematurely reveals all core results but without context. The results section requires much better guiding of the readers. The supporting information and its referencing in the main text are, unfortunately, very chaotic. Second, I have strong doubts about the correctness of the simulations regarding the N-H dissociation channel, as detailed below. I also have several minor issues that the authors hopefully can fix.

Response: We thank the reviewer for the attention of detail about the structure of the manuscript as well as the SI. We have made considerable changes to the manuscript in order to improve its readability, including, as suggested, the order of the presentation. We address the more specific comments about the text and the SI below.

Comment: Overall, I think that the manuscript - if the NH dissociation results are correct - is certainly of high interest and significance and it is great to see a new excited-state nonadiabatic method in action. Also the quality of the simulated spectra seem to be of very high quality. Nevertheless, while I would like to see the manuscript published, this is not possible in the current form.

Response: We again thank the reviewer for the positive evaluation of our work and its significance and interest to the community.

Comment: Major issues

1) The most severe problem of the manuscript is that the hydrogen dissociation process, which is already emphasized in the title, might be simulated in an inaccurate way. I first became suspicious about this in the description at the top of page 9 (related to Figure S14). The authors write that at 2.4fs, the S2 is the pipi^* state but at 4.0fs the S2 is the pisigma^* state. At 4.1fs, the population is transferred to the S1 state. Assuming that the wavepacket starts in the S2 state (never explicitly stated), it appears that over the course of less than 2fs, the electronic wave function evolves from pipi^* to pisigma^* , which would be extremely fast. My suspicion is rather that the

AIMS simulations only included S1 and S2, but not S3, as all population plots show only these two states (number of states is never given). If the S3 was included in the simulation, then at the S2/S3 conical intersection, a large fraction of the wavepacket should have transferred to the S3 in order to keep the pipi^* character. But if S3 is neglected, then all S2- \rightarrow S3 population transfers are automatically suppressed and the S2 population is forced to change character from pipi^* to pisigma^* in an unphysical way. The authors need to provide simulations of the dissociating initial conditions that include the S3 state. Only if the simulations including S3 also show strong population transfer to the pisigma^* , then the hydrogen dissociation mechanism can be trusted.

Response: We thank the reviewer for the attention to detail and for bringing to our attention this potentially important detail.

We have performed the simulations for the $\pi\sigma^*$ initial conditions with all three states explicitly included in the AIMS simulation and we find no changes in the population transfer. In particular, there is the same transfer of population to the trajectory basis functions that lead to N_1H dissociation. One of the two initial conditions gives identical results and transfers no population to S_3 . The other transfers 3% to S_3 . In both cases, as already noted, the trajectory basis functions that end in the $\pi\sigma^*$ dissociation channel have the same population with and without including S_3 in the dynamics simulation. This is now discussed in the revised text:

Additional calculations support the predicted $\pi\sigma^$ pathway. Both of the dissociative conditions display identical behavior when the S_3 state is included in the dynamics simulation, showing that its inclusion does not suppress the channel (see Supporting Information S6) [...]*

Comment: 2) There are multiple problems with the writing of the manuscript that could lead to confusion of the readers. From the flow of the text, one obtains the impression that the Results and Discussion section should actually start at the top of page 4. There, the first new results are already presented, in particular Figure 2, which shows the simulated O-edge XANES results that fit very well with experiment. Then, the authors tease the existence of the hydrogen dissociation channel at the end of the introduction. In the Results and Discussion, the authors start again at Figures 1 and 2, repeating information that was already written in the figure captions. I strongly suggest that the authors clean up the introduction to avoid repetition and to help readers separate previous work from the current work. Instead, the authors could use the space in the introduction to provide some essential information about the simulations, see my point 10) below. It might also be useful to formulate the goals of the study in the introduction. The rest of the Results and Discussion would also benefit strongly from better guides to the readers. Subheadings should be added to separate the discussion of electronic/spectroscopic results, nuclear motion results, and the new hydrogen dissociation channel.

Response: We thank the reviewer for pointing out the repetition in the text, and for the suggested changes to the introduction and the presentation of the results.

First, we have moved the start of Results and Discussion section as suggested, and made adjustments to the subsequent text in order to avoid repetition and to

present the results in an orderly fashion, which now begins with the pathways (Figure 1), followed by the $\pi\pi^* \rightarrow n\pi^*$ spectroscopic signature and the associated dynamics (Figures 2, 3, and 4), and, finally, by the discussion of the $\pi\sigma^*$ channel (Figure 5). With these adjustments, we do not think subheadings are needed.

Second, we have provided some additional information about the dynamics in the final paragraph of the introduction, as suggested. In particular, we now explicitly mention the simulation time (100 fs), the dynamics method (AIMS), and the excited state electronic structure method (EOM-CCSD/EOM-SCCSD):

Here we present the highest-level wavepacket simulation on the early dynamics of gas phase thymine to date. To the best of our knowledge, this is also the simulation with the highest level of electronic structure theory performed on a molecular system of this size. Thanks to recent developments [3-8], we were able to describe the electronic structure with the highly accurate coupled cluster singles and doubles (CCSD) [9] method in its equation of motion formulation for excited states (EOM-CCSD) [10]. This is the first time that this method is applied in nonadiabatic dynamics. Coupled cluster (CC) theory is well-known for effectively capturing dynamical correlation, but it has been widely regarded as unsuited for excited state dynamics due to the presence of numerical artifacts at conical intersections [3, 11-13]. Recent work has shown that these problems can be removed with similarity constrained CC (SCC) theory [4, 5, 8]. Here, we apply the SCC with singles and doubles (EOM-SCCSD) method to simulate the first 100 fs after photoexcitation using ab initio multiple spawning (AIMS) [14, 15], showing that it is possible to simulate nonadiabatic dynamics with a coupled cluster method that correctly describes conical intersections. This demonstrates that coupled cluster theory is a viable electronic structure method for simulating a range of photochemical processes.

Comment: 3) Closely connected with point 2), the authors need to make an effort to better organize and discuss their supplementary results. The supplementary information has 9 sections, 18 figures, and 4 tables. The only items that are mentioned in the main text are Sections S2, S1, S6, S7, and S9, in that order. Not a single supplementary figure or table is referenced. Even the few existing references are not very helpful, because within the supplementary information, referencing is also very bad. Section S1 mentions Figures S1-S5, and then the next reference is Figure S15 in Section S3. Figures S6-S12 are never mentioned (one can guess that they belong to Section S2). Figures S13 and S14 seem to belong to Section S6 (along with Figure S16). Figure S18 and Table S4 are never mentioned either (do they belong to Section S9?). Furthermore, the sections and figures/tables are badly aligned, for example does Figure S8 - belonging to Section S2 - only appear after Section S9. Moreover, there are 4 citations that only appear in the supplementary information, so they will not be indexed and their authors will not receive the appropriate credit. One of the citations is even to the authors' own Zenodo repository with additional simulation data, so the authors should have an interest that readers actually find this citation. I strongly recommend that the authors order the supplementary information correctly (please look up the latex command clearpage). All supplementary items (sections, figures, tables) should be referenced in the main text in the correct order and within the supplementary

information. I also suggest to add a table of content to the SI and to put all citations in the main text.

Response: We thank the reviewer for the useful suggestions. We have re-organized the SI in order to make the presentation of the results clearer. All figures and tables are now appropriately referenced within the SI. We have furthermore moved citations from the SI to the main text (to give appropriate credit) and we have made sure to reference all sections of the SI in the main text. We have decided not to reference all SI tables and figures in the main text as we believe such details are more appropriately referenced internally in the SI. The SI now has a table of contents with hyperlinks for easy navigation.

Comment: 4) Figure 1 is a very large figure that provides only very limited insight and appears almost decorative. One can observe that there are two potential energy surfaces, but the interesting crossing region is hardly visible due to the chosen colors and transparency. It is unclear whether this is a sketch or a real PES scan, and what the degrees of freedom are. It is also somewhat deceptive, because later the authors mention three essential degrees of freedom (r48, r56, and rNH), but Figure 1 gives the impression that there are only two. The orbital plots, which are in principle useful to introduce the electronic states to the readers, are rather small instead. As mentioned above, a further problem is that the caption already preempts several results from below, increasing the confusion whether the figure is a sketch or not.

Response: Figure 1 was meant as an illustration of the pathways. We have made various adjustments to address the comments. First, we now make sure to emphasize that the $\pi\sigma^*$ pathway follows a separate path on the potential energy surfaces, and we only illustrate the main pathway in the figure. Second, we have shortened the caption to avoid repetition and to leave certain details to the main text:

*Photochemical pathways in the **dynamics** simulation. Following photoexcitation to the bright $\pi\pi^*$ state (S_2), the simulation predicts two channels. The main channel is the $n\pi^*$ trapping channel. Here, the wavepacket passes through the S_1/S_2 intersection and heads toward an $n\pi^*$ minimum on the S_1 surface. This minimum is reached in two ways, either by heading to the minimum directly (solid line) or by reaching it indirectly through a $\pi\pi^*$ region on S_1 (dashed line). The second channel is an N-H dissociation channel, **which follows a separate pathway on the potential energy surfaces (not shown). Here the wavepacket on S_2 moves to regions with $\pi\sigma^*$ character near a degeneracy with S_3 and S_1 at extended N-H bond lengths, followed by transfer to S_1 and dissociation of one of the N-H bonds. The three states involved ($n\pi^*$, $\pi\pi^*$, and $\pi\sigma^*$) are illustrated by corresponding natural transition orbitals.***

We have also made the orbital plots bigger to make it easier to visualize the electronic states involved in the two pathways. We have furthermore removed (as suggested by another reviewer) the notation $S_1(n\pi^*)$ and $S_2(\pi\pi^*)$ in the figure and the caption to make it clear that the state character is not associated with the energy ordering.

Comment: 5) Figure 3 appears somewhat out of place in the main manuscript. In contrast to the other main figures, Figure 3 does not provide any insight into the photodynamics of thymine. Given how the title, abstract, and introduction lead the

readers into the paper, the figure therefore does not fit into the narrative. Possibly even the authors have realized this, because the Figure is only mentioned very briefly once: “Close to the intersection, CCSD exhibits numerical artifacts which can be removed with SCCSD. Figure 3 shows one of these artifacts as well as how it is corrected.” While I can understand the authors that they want to present the strengths of the new method, I do not have the impression that this figure is the best way to do so. It also does not help that the SCCSD method was actually only used for 2/16 simulations, whereas CCSD was used for the remainder. They should either expand the discussion of the figure (e.g., in a separate subheading at the end of the Results and Discussion) or move the figure to the SI.

Response: We thank the reviewer for the suggestion, but we have decided to keep the figure in the manuscript. An important part of the work is the methodological advance that made the dynamics simulations feasible. We therefore believe it should be highlighted in the main text. We have expanded the discussion, as suggested:

Close to the intersection, CCSD exhibits numerical artifacts (unphysical complex energies and distorted potential energy surfaces) which can be removed with SCCSD. Figure 3 details a simulated initial condition that encounters such artifacts and shows how the issue is averted with the SCCSD method.

Comment: 6) One of the striking results of the new study is the simulated time-resolved O-edge XANES in Figure 2. It lends significant credibility to the results of the study (albeit not about the NH dissociation). Because of this, I find it a pity that the authors did not apply the same level of attention to the N-edge XANES, instead hiding it in the supporting information. If the NH dissociation channel is real and the authors want to stimulate further experiments on thymine, then it would only be consequential to simulate the time-resolved N-edge XANES and show it in the main text.

Response: We thank the reviewer for the suggestion but we think further simulations are not warranted in this case. The N-edge calculations in the SI already provide evidence that the $\pi\sigma^*$ channel would have a distinct signature in an N-edge XANES. It should also be noted simulating the time-resolved N-edge XANES would require extensively sampling the $\pi\sigma^*$ channel. This would surely be interesting work, but we believe it is beyond the scope of the current study.

On the point about what to highlight in the main text, this is naturally a difficult task in a study with many results. We believe that the major findings are well covered by Figures 1–5 and that the manuscript would not benefit from a further expansion.

Comment: Minor issues

7) Abstract and introduction: The authors currently do not mention in any place whether they talk about the dynamics of thymine in gas-phase, in solution, or in some biological environment. Only readers that are aware of the experimental methods or that know the citations can conclude that the manuscript is about gas-phase dynamics. This information is critical, because thymine dynamics is significantly different in different environments. Especially N-H dissociation will be very different in condensed

phase. Please add appropriate hints in the abstract and introduction (and possibly the title).

Response: We thank the reviewer for pointing this out. We agree with the reviewer that it would be beneficial for the reader to make explicit that the study is about gas-phase dynamics. The abstract now reads

The fate of thymine upon excitation by ultraviolet radiation has been the subject of intense debate over the past three decades. Today, it is widely believed that its ultrafast excited state decay in the gas phase stems from a radiationless transition from the bright $\pi\pi^$ state to a dark $n\pi^*$ state. However, conflicting theoretical predictions have made the experimental data difficult to interpret. Here we simulate the gas phase ultrafast dynamics in thymine at the highest level of theory to date, performing wavepacket dynamics with a new coupled cluster method. Our simulation confirms an ultrafast $\pi\pi^*$ to $n\pi^*$ transition ($\tau = 41 \pm 14$ fs). Furthermore, the predicted oxygen-edge X-ray absorption spectra agree quantitatively with the experimental results. Our simulation also predicts an as-yet uncharacterized photochemical pathway: a $\pi\sigma^*$ channel that leads to hydrogen dissociation at one of the two N-H bonds in thymine. Similar behavior has been identified in other heteroaromatic compounds, including adenine, and several authors have speculated that a similar pathway may exist in thymine. However, this was never confirmed theoretically or experimentally. This prediction calls for renewed efforts to experimentally identify or exclude the presence of this channel.*

While the introduction reads

Thymine, like other nucleobases, undergoes ultrafast radiationless relaxation back to the ground state after being excited by ultraviolet radiation. This property has been tied to the resilience of genetic material against light-induced damage [21]. However, the exact mechanism of this decay is not fully understood and has been a subject of debate for several decades. Gas phase experiments have identified at least two excited state decay channels, one with a lifetime on the order of $\lesssim 100$ fs, and one considerably longer, on the order of several ps [22–25]. Yet, the underlying mechanisms have been challenging to discern, with proposed explanations necessarily relying on simulations of the molecular dynamics. These simulations, in turn, introduce approximations with errors that are difficult to control. Different theoretical methods have therefore produced different explanations, and a consensus has yet to emerge.

[...]

Here we present the highest-level wavepacket simulation on the early dynamics of gas phase thymine to date.[...]

Comment: 8) Introduction: "Different theoretical methods have therefore produced different explanations, and a consensus has yet to emerge." Are the authors here referring to the next paragraph? They might want to add the citations already to this sentence, to make sure what they are referring to. Furthermore, a statement could be added that most of the non-consensus is due to different electronic structure levels of theory.

Response: We have adjusted the sentence to put emphasis on the central role of the electronic structure approximation (and hence the next paragraph):

These simulations, in turn, introduce approximations with errors that are difficult to control. Different theoretical methods, and in particular different electronic structure methods, have therefore produced different explanations, and a consensus has yet to emerge.

Comment: 9) While paragraphs 2 and 3 nicely summarize a lot of experimental and theoretical evidence on thymine, they completely spare out intersystem crossing. This probably does not meaningfully affect the simulations, but for completeness the few-% triplet yields in thymine should be mentioned.

Response: We thank the reviewer for the suggestion. Intersystem crossing is now mentioned in the introduction for completeness:

After passing through the $\pi\pi^/n\pi^*$ conical intersection, parts of the wavepacket appears to get trapped in a minimum on the $n\pi^*$ surface. On the timescale of a few ps, there is evidence that thymine further undergoes intersystem crossing from the $n\pi^*$ singlet to a $\pi\pi^*$ triplet [17] [...]*

Comment: 10) Although it is fine that the bulk of the computational details is collected in the Methods section at the end, the most important pieces of information should also be given in the main text, else the reader will constantly need to leave back and forth. Some important details that should be added to the beginning of the Results and Discussion (or introduction) are the following. (i) The simulations are actually performed with EOM-CCSD/EOM-SCCSD. Most readers will probably assume ground state calculations if only "CCSD" is written, or will wonder what excited-state theory is used (response theory like in CC2, EOM, delta-CC, etc). (ii) That the nonadiabatic dynamics theory is multiple spawning. (iii) That the simulation time was 100fs. In this context, one should also mention that even SCCSD cannot describe S1/S0 conical intersections, so longer simulation times would not have been useful.

Response: We thank the reviewer for the suggestion and we agree that adding these details to the main text would improve the readability of the manuscript. These details are now provided in the last paragraph of the introduction:

Here we present the highest-level wavepacket simulation on the early dynamics of gas phase thymine to date. To the best of our knowledge, this is also the simulation with the highest level of electronic structure theory performed on a molecular system of this size. Thanks to recent developments [3–8], we were able to describe the electronic structure with the highly accurate coupled cluster singles and doubles (CCSD) [9] method in its equation of motion formulation for excited states (EOM-CCSD) [10]. This is the first time that this method is applied in nonadiabatic dynamics. Coupled cluster (CC) theory is well-known for effectively capturing dynamical correlation, but it has been widely regarded as unsuited for excited state dynamics due to the presence of numerical artifacts at conical intersections [3, 11–13]. Recent work has shown that these problems can be removed with similarity constrained CC (SCC) theory [4, 5, 8]. Here, we apply the SCC with singles and doubles (EOM-SCCSD) method to simulate the first 100 fs after photoexcitation using ab initio multiple spawning (AIMS) [14, 15], showing that it is possible to simulate nonadiabatic dynamics with a coupled cluster method that correctly describes conical intersections. This demonstrates

that *coupled cluster theory* is a viable electronic structure method for simulating a range of photochemical processes.

Comment: 11) The percentages in the caption of Figure 1 indicate that only two pathways are present, trapping in $n\pi^*$ (87%) and NH dissociation (13%). This finding requires some contextualization, as many previous studies have found/indicated that a large fraction of population goes directly from the $\pi\pi^*$ to the ground state, with a short stretch where the $\pi\pi^*$ is S_1 .

Response: We now mention this specifically when we introduce Figure 1:

The main photochemical pathways in our simulation are illustrated in Figure 1. First, our simulation confirms an ultrafast $\pi\pi^$ to $n\pi^*$ conversion. We find no significant $\pi\pi^*$ trapping and no direct $\pi\pi^*$ relaxation to the ground state. [...]*

We also state in the introduction that there are studies that find direct $\pi\pi^*$ to ground state relaxation, which should provide readers with the required context:

[...] These simulations disagree, however, on the amount of $\pi\pi^$ trapping, as well as the timescale and nature of the subsequent processes, with proposed mechanisms including $\pi\pi^*$ to $n\pi^*$ relaxation [16] and direct $\pi\pi^*$ relaxation to the ground state [26] [...]*

Comment: 12) "Moreover, we find frequency oscillations" What does this mean? Should it simply read "we find oscillations" or rather "we find oscillations in the frequency of some signal"?

Response: We were referring to the oscillations in the signal in Figure 2 (Panel B) at around 526 eV. We have adjusted the text as suggested:

[...] Moreover, we find oscillations in the signal at 526 eV associated with dynamics on the $n\pi^$ state. These oscillations have a period of about 20 fs and an amplitude of about 1 eV (see panel B) [...]*

Comment: 13) The discussion in the top paragraph of page 7 about the difference between adiabatic and diabatic populations (and related time constants) is not really new anymore. There are many works in the literature that show that adiabatic populations are not useful to compare to experimental observables. Maybe the discussion could be rephrased such that readers do not get the impression that this aspect is new. Additionally, it might be useful to show the electronic populations (any) in the main manuscript, maybe in Figure 2.

Response: We agree with the reviewer and we have shortened the paragraph to make this discussion more focused and to not give an impression that this is new:

The $\pi\pi^/n\pi^*$ conversion time of $\tau = 41 \pm 14$ fs was determined by analyzing the growth of the 526 eV signal in the simulated spectrum. This time constant is consistent with the rate of $\pi\pi^*/n\pi^*$ conversion in the simulated dynamics, that is, from the observed change in electronic character from $\pi\pi^*$ to $n\pi^*$. We find a rapid adiabatic population transfer from S_2 to S_1 ($\tau = 17 \pm 1$ fs) in our simulation. However, when the adiabatic states are decomposed*

into their diabatic components, and in particular into their $\pi\pi^$ and $n\pi^*$ components, we see that the growth in the $n\pi^*$ character ($\tau = 37 \pm 9$ fs) is in close agreement with the time constant determined from the simulated spectrum ($\tau = 41 \pm 14$ fs). This shows that the 526 eV signal in the spectrum is due to the electronic $n\pi^*$ character.*

We thank the reviewer for the suggestion to show the populations in the main text, but we have decided against it as we believe it would make the manuscript less concise and that it would not add significant value as part of the main text.

Comment: 14) In the caption of Figure 3: "Dynamics with CCSD (panels A and B) and SCCSD (panels C and D)." I think this caption is not appropriate. Panels A and C do not show dynamics, but a plot of a conical intersection. The fragment "branching or gh plane for an initial condition" is even more confusing, because an initial condition does not have a branching plane.

Response: We thank the reviewer for pointing out that the original caption in Figure 3 needed to be made more precise. We have modified it to make it more clear:

An example of a numerical artifact encountered during the dynamics with EOM-CCSD (panels A and B) and their correction with EOM-SCCSD (panels C and D). Panels A and C show a branching or gh plane for a conical intersection encountered during the simulation of one of the initial conditions. ΔE is the energy difference between S_1 and S_2 . For CCSD (A), there is an ellipse of degeneracy in the gh plane with unphysical complex-valued energies in the interior of the elliptical boundary; for SCCSD (C), we instead see a single point of degeneracy and no unphysical energies. In CCSD simulations, the wavepacket may approach the intersection too closely and end up in the region with complex-valued energies (the interior of the ellipse in the gh plane). Whenever this happens, we re-run the simulation with SCCSD. Panels B and D show the corresponding potential energy curves for the center of a nuclear trajectory basis function with CCSD (B) and the same trajectory basis function with SCCSD (D). At 71 fs, the CCSD simulation enters the complex-valued region and is terminated (B). The SCCSD simulation, on the other hand, does not encounter any problems (D). The expansion point used in the branching plane calculation is the geometry with the smallest ΔE (as given by SCCSD) in the nonadiabatic event at 71 fs.

Comment: 15) In Figure 4, it is somewhat hard to see how the wavepacket is actually moving. I recommend to add some arrows to guide the eye. Furthermore, please indicate whether the total wavepacket is shown or only the one on the S2 surface (or S1?). I am also not really sure what the message of this figure should be. The main text discussed the figure only briefly in a short paragraph on page 7. Maybe this could be better worked out, including a comparison to other literature works?

Response: It is the total wavepacket. We have revised the caption:

Total nuclear density snapshots in the C_5-C_6 and C_4-O_8 coordinates. Three important stationary points are shown [...]

The caption guides the reader through the various snapshots one by one, which we believe explain various steps observed in the dynamics:

[...] The wavepacket quickly moves away from the Franck-Condon region (S_0^{min}) and towards the minimum-energy conical intersection, where it starts transferring population to the $n\pi^*$ surface (19 fs), eventually causing the wavepacket to split (38 fs). At longer times, after almost all of its population has transferred to the lower surface (58 fs), the wavepacket settles in the vicinity of a minimum on the $n\pi^*$ surface (77 and 96 fs).

The figure is already rather busy and we have decided against adding arrows to guide the eye.

The purpose of the figure is to show that the two-step mechanism proposed by Wolf et al. [20] is also observed in the simulated dynamics. As we write in the main text:

By analyzing stationary points on S_1 and S_2 , Wolf et al. [20] suggested that the excited state decay of thymine follows a two-step process in the C_5-C_6 and C_4-O_8 coordinates (see Figure 2A for atom labelling): following photoexcitation, the C_5-C_6 bond is first elongated, and along this stretching coordinate, the S_1/S_2 intersection seam is accessible; then, after interconversion to the S_1 state, the C_4-O_8 bond elongates as the wavepacket heads towards the $n\pi^$ minimum on S_1 . This picture is borne out by our dynamics simulation. In Figure 4, we show the time evolution of the nuclear density in the C_5-C_6 and C_4-O_8 bond coordinates, and we indeed see this two-step process unfolding in real time [...]*

Comment: 16) In Figure 5, I was wondering why the authors employ a very high level of theory for the absorption spectrum (CC3 with a basis set tailored for Rydberg states) but then simply obtain the spectrum by Gaussian convolution, where they did not even match the Gaussian width to the first absorption band. Why was the spectrum not computed for the Wigner distribution of initial conditions? In this way, agreement with experiment could be strongly improved.

Response: This high-level calculation was done to accurately estimate the second $\pi\pi^*$ absorption band, as this would potentially impact the interpretation of short-wavelength experiments aimed at detecting $\pi\sigma^*$ -mediated NH dissociation. In the main text, we write:

[...] However, wavelengths in the range 230–200 nm may not be directly comparable to our simulations as it would also excite the system to states above the $S_2(\pi\pi^)$ state, in particular, to the $\pi\pi^*$ band that lies about 1.0 eV above $S_2(\pi\pi^*)$ (see Figure 5E and Supporting Information S5). This is similar to adenine, where several states may contribute to the $\pi\sigma^*$ dissociation channel [27] [...]*

We now also provide the UV-vis spectrum for the Wigner distribution at the CCSD level of theory in Supporting Information S5.

Comment: 17) Conclusion: In my opinion, "with the highest level of electronic structure" is a controversial statement here. Many researchers will disagree that a method that cannot describe S_1 - \rightarrow - S_0 dynamics is "the highest level" for a system like thymine, where this dynamics is very important. My point here is that excited-state electronic structure methods are too diverse that one can simplify them into a one-dimensional scheme where one is clearly the "highest". Hence, I suggest to rephrase. Also note that other high-level nonadiabatic dynamics for nucleobases have been reported before, e.g.,

Response: We agree with the reviewer that an evaluation of the accuracy of a method must be considered in the context of the system it is describing, which in this case is the early-time dynamics of thymine. This means of course that one cannot state that a method is “highest level” without this context.

With this in mind, we maintain that a simulation of the early excited state dynamics in thymine with a full description of double excitations is appropriately said to be a higher level of theory than in previous studies. Here we also note that another reviewer agreed with this statement. This is both because it incorporates double excitations non-perturbatively (thereby exceeding the description of ADC(2) and DFT) and because the early dynamics in thymine is known to be rather sensitive to dynamical correlation, making CAS-methods less reliable. Moreover, the targeted parts of the dynamics, in particular, the ultrafast $\pi\pi^*$ to $n\pi^*$ dynamics and the involvement of the dissociative $\pi\sigma^*$ state, is within what is expected to be the valid application domain of the EOM-CCSD method.

Of course, there are systems and processes where other methods may exceed the capabilities of standard EOM-CCSD. We now provide further remarks in this direction in the conclusion:

All electronic structure methods have limitations and this is also true for coupled cluster theory. As a single-reference method, it does not give a proper description of conical intersections with the ground state, and high-order treatments are needed to describe excited states with significant double excitation character. Nevertheless, given their ability to capture dynamical correlation and to provide systematically improvable predictions, we expect that coupled cluster methods will shed new light on the photochemistry of several important systems.

Reviewer 3

Comment: The manuscript reports a theoretical study of the ultrafast dynamics of isolated thymine molecules following photoexcitation at UV energies, using wavepacket dynamics together with a new coupled cluster method. The excited state dynamics of isolated thymine molecules have been studied previously, but I am willing to accept the authors assertion that this study has been undertaken at the highest level of theory to date. The study identifies that initial photoexcitation is to the ‘bright’ S2 state, which arises via a $\pi^* \leftarrow \pi$ promotion. The S1 state has predominant $n\pi^*$ character in the vertical region, but photoexcited molecules rapidly sample the region of conical intersection (CI) between the S2 and S1 states and most of the initial S2 population transfers to the S1 state at this CI. This photophysical interpretation, and the timescale predicted by the present simulation are broadly consistent with the results of prior work. The predicted time-resolved X-ray absorption spectrum of thymine following ultrafast UV photoexcitation, measured at the oxygen-edge, also matches a prior experimental measurement; the signature of the emerging and (over a short timescale) persistent $n\pi^*$ population is clearly revealed. The results described thus far

provide little fundamental new knowledge, but certainly reinforce and validate prior discussions. However, the present calculations also reveal an additional (minor) decay pathway, involving H atom loss via breaking of the N1-H bond. Activity in such a channel might not be viewed as surprising, at least by those familiar with other the photophysics of other N-containing heterocycles, but the present work provides direct evidence that it may play a role.

The submitted manuscript is very readable, demonstrates use of a very recent and improved computational method for studying excited state photophysics in medium sized (i.e. 10 (light) atom) molecules, confirms much of the current early-time photochemical knowledge for this molecule and identifies some activity in a hitherto unreported dissociation pathway. As such, I'm content that it passes the suitability threshold for publication in Nature Communications. I now offer suggestions as to how (in my opinion) the manuscript could be improved, but the authors should feel free to decide which (if any) they choose to act on.

Response: We thank the reviewer for the positive evaluation of our work's readability, significance, and suitability for the journal.

Comment: In several places I felt that the authors were trying a bit too hard to 'sell' their work. The title provides a good case in point. Hydrogen does not 'dissociate' – it's an N–H bond that dissociates. (The phrase hydrogen dissociation also appears elsewhere in the manuscript). I would also quibble with the use of the word 'Unexpected'. 'Hitherto unreported' might be fairer. As the authors discuss later in the manuscript, analogy with other N-heterocycles implies that there will be (more than) one dissociative $(n/\pi)\sigma^*$ potential that could support N–H bond fission. The process has not been reported hitherto in thymine but, again, the authors rightly point out that this may just be because the right type of experiment has yet been performed at suitably high excitation energies. I'm also not fond of claims to be the 'first' to do something – as claimed twice (for essentially the same point) in in the paragraph leading into Fig 1 and reiterated again in the concluding paragraph.

Response: We thank the reviewer for the comments. With respect to the title and the phrase "hydrogen dissociation", we believe it is clear that it refers to the loss of hydrogen in this context. This meaning is also clear from the abstract:

[...] Our simulation also predicts an as-yet uncharacterized photochemical pathway: a $\pi\sigma^$ channel that leads to hydrogen dissociation at one of the two N-H bonds in thymine [...]*

We do not think that replacing "hydrogen dissociation" with "N-H dissociation" would make a significant difference here.

The N-H dissociation is "unexpected" for two reasons. The first is that it has not been identified in other simulations on thymine, and that discussions have, as a result, centered on the $\pi\pi^*$ and $n\pi^*$ states. The second is that there is experimental evidence to the contrary. It is true that the N-H dissociation fits into the wider context of $\pi\sigma^*$ states in other N-heterocycles, as we also discuss at length in the manuscript. This does not mean, however, that the finding is therefore not "unexpected".

We thank the reviewer for pointing out unnecessary repetitions. First, we have adjusted the paragraph in the introduction, which now reads:

Here we present the highest-level wavepacket simulation on *the early dynamics of gas phase thymine* to date. To the best of our knowledge, this is also the simulation with the highest level of electronic structure theory performed on a molecular system of this size. Thanks to recent developments [3–8], we were able to describe the electronic structure with the highly accurate coupled cluster singles and doubles (CCSD) [9] method *in its equation of motion formulation for excited states (EOM-CCSD)* [10]. This is the first time that this method is applied in nonadiabatic dynamics. Coupled cluster (CC) theory is well-known for effectively capturing dynamical correlation, but it has been widely regarded as unsuited for excited state dynamics due to the presence of numerical artifacts at conical intersections [3, 11–13]. Recent work has shown that these problems can be removed with similarity constrained CC (SCC) theory [4, 5, 8]. Here, we apply the SCC with singles and doubles (*EOM-SCCSD*) method to *simulate the first 100 fs after photoexcitation using ab initio multiple spawning (AIMS)* [14, 15], showing that it is possible to simulate nonadiabatic dynamics with a coupled cluster method that correctly describes conical intersections. This demonstrates that *coupled cluster theory* is a viable electronic structure method for simulating a range of photochemical processes.

We have also adjusted the conclusions to remove the pointed-out repetition:

This work was made possible by recent developments in coupled cluster theory. Earlier work by some of the authors [3–5] had already indicated that the method could be modified to correctly describe conical intersections and therefore also nonadiabatic dynamics. Here, we have shown that the method introduced in these papers, the similarity constrained coupled cluster method, can in fact be applied in nonadiabatic dynamics simulations, opening up a range of applications that may now be studied with the hierarchy of coupled cluster methods.

Nevertheless, we have kept one reference to being “first” in the introduction. In this context, we emphasize that the challenge of using the coupled cluster method in nonadiabatic dynamics has been a long-standing theoretical issue in the community [3–5, 7, 8, 11, 12]. There is therefore considerable significance, in our view, in our ability to perform this kind of dynamics simulation for the first time. This context is therefore still provided in the revised text:

This is the first time that this method is applied in nonadiabatic dynamics. Coupled cluster (CC) theory is well-known for effectively capturing dynamical correlation, but it has been widely regarded as unsuited for excited state dynamics due to the presence of numerical artifacts at conical intersections [2, 10–12] [...]

Comment: Generally interested readers might welcome a bit more preamble. The Introductory paragraph starts with ‘Thymine, like other nucleobases, undergoes ultrafast radiationless relaxation back to the ground state after being excited by ultraviolet radiation. This property has been tied to the resilience of genetic material against light-induced damage [1]. However, the exact mechanism of this decay is not fully understood and has been a subject of debate for several decades’ – all of which is true. Most readers (i.e. those outside the ultrafast, first 100 fs, isolated molecule community), however, might then be disappointed to find that the ‘new insights’ reported here never actually get to the ‘back to the ground state’ bit. For general interest, it might be worth being clearer that this work only addresses the earliest time dynamics in isolated thymine molecules. It is silent on how excited thymine molecules in the n* state couple back to the ground state, or how they would decompose further in the

absence of collisions. The ‘resilience of genetic material against light-induced damage’ applies when molecules such as thymine are in a condensed phase environment, and the energy introduced by photon absorption can be dissipated to the background.

Response: We thank the reviewer for pointing out that the focus of the study, and in particular to the first 100 fs and to the gas phase, was not sufficiently emphasized. First, in the abstract, we have added “early gas phase” to clarify:

*[...] Here we simulate the **early gas phase** ultrafast dynamics [...] Our simulation confirms an ultrafast $\pi\pi^*$ to $n\pi^*$ transition ($\tau = 41 \pm 14$ fs) [...]*

In the introduction, we now also specify that we are considering the first 100 fs:

*[...] Here we present the highest-level wavepacket simulation on **the early dynamics of gas phase thymine** to date [...] Here, we apply the SCC with singles and doubles (EOM-SCCSD) method to **simulate the first 100 fs after photoexcitation using ab initio multiple spawning (AIMS)** [14, 15], showing [...]*

Note that we here also mention the methods explicitly (EOM-SCCSD and AIMS), as another reviewer requested more methodological details in the Introduction.

We agree that other aspects of the dynamics in thymine is of high interest, such as modeling its longer-time dynamics as well as the effects of the environment. However, given the amount of background required to understand investigations of the early gas phase dynamics, we think an expanded discussion of such aspects is outside the scope of the manuscript and that readers may instead consult the literature (e.g., Ref. 1).

Comment: Readers might also appreciate a bit more clarity about the relevant excitation energies/wavelengths. The Wolf et al results (ref 16) against which the present theoretical data are compared were obtained following excitation with a UV pulse centred at a wavelength of 267 nm (i.e. the red wing of the parent absorption spectrum). I did not find an explicit statement in the present manuscript as to the excitation energy used in the simulations. (I may have missed it). But, from Fig. 5, I assume it was the energy appropriate for vertical excitation to the calculated S2 state – i.e. a photon energy 0.5 eV higher than used in the Wolf et al. experiments. This discrepancy raises several minor worries in my mind. Should one be surprised that there is still a 0.2 eV mismatch between the maxima in the long wavelength feature in the experimental and computed spectra of thymine, given the assertion that the paper reports ‘the highest level of electronic structure theory performed on a molecular system of this size’? The dynamical simulations return decay lifetimes and channel branching ratios based on the calculated potentials and assumed excitation energy. The 267 nm experiments presumably sample the S2 potential at a slightly different starting geometry (away from vertical) and at a different total energy. To what extent should one expect the measured and predicted timescales to be the same? The manuscript currently emphasises the good agreement, but nowhere discusses the extent to which the two studies are (or are not) exploring the same thing. The calculated branching ratios are reported to the nearest percent, which is fair enough; that’s what the simulations return. But as Fig 5 shows, the calculations still don’t get the S2 state energy correct. Do the authors have a feel for how the estimated branching into the N-H bond fission channel depends on the reliability of their $\pi\pi^*$ and $\pi\sigma^*$ excitation energies along Q

in Fig 4B (the cut along the $\pi\sigma^*$ potential at large Q looks very questionable) or to the excitation energy assumed? In similar vein, the current manuscript contains the sentence on ‘Other experiments have appeared to disconfirm the existence of a dissociative $\pi\sigma^*$ channel’ but gives no reference(s) indicating to which other experiments the statement applies.

Response: The reviewer is correct to point out that the simulation may not coincide exactly with the experiment. For example, the 267 nm pulse may introduce a bias to certain parts of the $\pi\pi^*$ absorption band. In our dynamics, we sample the entire band instead of restricting the excitations to a window (as is sometimes done). Although it is difficult to determine the precise effect of this (and we have decided not to speculate), we note that all of the sampled initial conditions move quickly and in tandem towards the S_2/S_1 MECI in the first 20 fs (see Figure 4), suggesting that there the variation in the exhibited $\pi\pi^* \rightarrow n\pi^*$ dynamics in the sampled initial conditions is small.

With respect to the deviation from the experimental peak in Figure 5, we emphasize that this is expected due to several factors (there is no applied shift, it is the vertical excitation energy, and systematic shifts of peaks typically occur with different descriptions of the correlation and the basis). With respect to the relevance of this, we also emphasize that there is no direct relation between the peak position and the reliability of the excited state dynamics, which depend on a combination of factors, including the potential energy landscapes, the crossing regions, and the relative $S_1/S_2/S_3$ energies over extended regions of the surfaces. The spectrum was simulated to target the separation between the first and second $\pi\pi^*$ bands, which should be well represented by EOM-CC3 and may affect the interpretation of short-wavelength TKER spectra:

[...] leads us to suggest that if the TKER experiments for thymine described by Schneider and coworkers [1] are carried out using excitation wavelengths in the range 230–200 nm (as also suggested in Ref. 28), then one might observe anisotropic, fast H-atom peaks consistent with this dissociative channel. However, wavelengths in the range 230–200 nm may not be directly comparable to our simulations as it would also excite the system to states above the $S_2(\pi\pi^)$ state, in particular, to the $\pi\pi^*$ band that lies about 1.0 eV above $S_2(\pi\pi^*)$ (see Figure 5E and Supporting Information S5) [...]*

With respect to the N-H scan, we now provide further calculations:

[...] while the $\pi\sigma^$ state has Rydberg character in the Franck-Condon region, this character decreases as the N_1 -H bond extends and the state becomes involved in the dynamics. Potential energy curves along the N_1 -H bond with a diffuse basis set (aug-cc-pVDZ) and with a higher-order correlation treatment (coupled cluster with perturbative triples, CC3 [2]) suggest that the pathway is still present with more accurate treatments (see Supporting Information S7).*

With respect to the missing reference, we have revised the text to make it clear to which reference the statement about other experiments was made:

[...] Some experimental data have appeared to disconfirm the existence of a dissociative $\pi\sigma^$ channel, as reported TKER spectra showed only smooth, isotropic H-atom kinetic energy, indicating no involvement of ultrafast, $\pi\sigma^*$ -mediated N-H dissociation [1] [...]*

Comment: The last sentence in the caption to Fig. 5 ‘The shaded area corresponds to wavelengths where TKER spectra have not been recorded’ is rather misleading. TKER spectra have not been recorded at any shorter wavelengths on the right-hand side of that figure either. More sensible might be to use a shaded region to show where TKER measurements have been reported.

Response: We thank the reviewer for the suggestion, but we have decided to keep the shaded area in order to highlight where experimentalists have suggested to perform further TKER experiments. This is now mentioned explicitly:

[...] This behavior for adenine, coupled with the presence of the $\pi\sigma^$ N-H dissociation in our dynamics, leads us to suggest that if the TKER experiments for thymine described by Schneider and coworkers [1] are carried out using excitation wavelengths in the range 230–200 nm (as also suggested in Ref. 28), then one might observe anisotropic, fast H-atom peaks consistent with this dissociative channel. However, wavelengths in the range 230–200 nm may not [...]*

Comment: I have not checked the references, but it is obvious that refs 16, 18, 19, 27, 43 and 45 are all missing key details.

Response: We thank the reviewer for pointing this out and we have updated Refs. 16, 18, 19, 27, 43, and 45.

Reviewer 4

Comment: The manuscript the first ab initio on-the-fly simulation of dynamics through conical intersections utilizing the coupled-cluster singles and doubles (CCSD) method. The article addresses and clarifies a long-standing controversy surrounding the dynamics following excitation of the lowest $1p\pi^*$ state of thymine. Furthermore, the study confirms, for the first time via computation, the existence of a competing relaxation channel via the lowest $1p\sigma^*$ state for a pyrimidine base. This significant discovery not only advances our understanding of pyrimidine base behavior but also broadens our comprehension of electronic structure and dynamics in molecular systems.

In summary, the manuscript represents a methodologically robust and intellectually stimulating contribution to the field of computational chemistry. The significance of its findings, combined with the rigorous computational approach employed, makes it an excellent fit for publication in Nature Communications. Therefore, I strongly

recommend publication of the paper if the following points are clarified by the authors.

Response: We thank the reviewer for the positive evaluation of the significance of the work and its suitability for the journal.

Comment: (1) Fig. 1 is not correct in this form. Either the potential energy surfaces (PES) of a three-state intersection (S_1 , S_2 , S_3) should be shown, or the $\pi\sigma^*$ channel should be removed. The intersections between three states are clearly seen in Fig. 5E.

Response: We thank the reviewer for pointing this out and we have made several adjustments to the figure. First, we have removed the attachment of electronic character ($n\pi^*$, $\pi\pi^*$, $\pi\sigma^*$) labels to the adiabatic states (S_1 , S_2 , S_3) in the figure and the caption. We have also made adjustments to throughout the main text to address this issue. Second, the potential energy surfaces now only illustrates the $\pi\pi^* \rightarrow n\pi^*$ pathway and this is made clear in the figure and the caption. The revised caption reads:

*Photochemical pathways in the **dynamics** simulation. Following photoexcitation to the bright $\pi\pi^*$ state (S_2), the simulation predicts two channels. The main channel is the $n\pi^*$ trapping channel. Here, the wavepacket passes through the S_1/S_2 intersection and heads toward an $n\pi^*$ minimum on the S_1 surface. This minimum is reached in two ways, either by heading to the minimum directly (solid line) or by reaching it indirectly through a $\pi\pi^*$ region on S_1 (dashed line). The second channel is an N-H dissociation channel, **which follows a separate pathway on the potential energy surfaces (not shown)**. Here the wavepacket on S_2 moves to regions with $\pi\sigma^*$ character near a degeneracy with S_3 and S_1 at extended N-H bond lengths, followed by transfer to S_1 and dissociation of one of the N-H bonds. The three states involved ($n\pi^*$, $\pi\pi^*$, and $\pi\sigma^*$) are illustrated by corresponding natural transition orbitals.*

Note that the $\pi\sigma^*$ state is illustrated (with the natural transition orbitals) but that the pathway is no longer illustrated in the figure, which is also emphasized in the revised caption:

*The second channel is an N-H dissociation channel, **which follows a separate pathway on the potential energy surfaces (not shown)** [...]*

Comment: (2) The discussion of the nonadiabatic dynamics, especially on p. 7, is confusing. The notation $S_2(\pi\pi^*)$, $S_1(n\pi^*)$ and $S_3(\pi\sigma^*)$ does not make sense when there are multiple intersections of adiabatic PES in the vicinity of the Franck-Condon (FC) zone, e.g. in Fig. 1. Adiabatic surfaces are defined as S_1 , S_2 , S_3 and are ordered by energy everywhere. Diabatic surfaces are defined as $1\pi\pi^*$, $1n\pi^*$,

1p σ * and are not ordered by energy. Adiabatic surface S_n can have 1p π i*, 1n π i* or 1p σ * electronic character, depending on the nuclear geometry.

Response: We thank the reviewer for pointing this out. We have adjusted the text (generally but also on p. 7) to more clearly separate the adiabatic surfaces from the electronic characters.

Comment: (3) The formulation on p. 7: “the adiabatic populations overestimate the true rate of internal conversion” turns the logic on the head. The molecular observables are the electronic populations and the time derivatives of the adiabatic populations are the rates. Depending on the situation (PES, transition dipole moments, type of signal), the time-resolved signal may approximately report adiabatic populations, diabatic populations or none of them. Adiabatic and diabatic populations are related by a geometry-dependent unitary transformation, as has been discussed since decades in the quantum wavepacket literature and more recently also in on-the-fly surface-hopping dynamics, e.g. Zhou et al., JPCLet. 2019, 10, 7062 or Xie et al., JCP 150, 154119 (2019). The discussion in Section S1 of the SI should be embedded in the existing literature.

Response: We thank the reviewer for pointing out that this discussion could be improved. We have rewritten the paragraph in question to make it clear that there is no expected connection between the adiabatic populations and the signal, and that in this case (where we are considering transition strengths in the X-ray absorption spectrum) the $n\pi^*$ character is the relevant variable:

The $\pi\pi^/n\pi^*$ conversion time of $\tau = 41 \pm 14$ fs was determined by analyzing the growth of the 526 eV signal in the simulated spectrum. This time constant is consistent with the rate of $\pi\pi^*/n\pi^*$ conversion in the simulated dynamics, that is, from the observed change in electronic character from $\pi\pi^*$ to $n\pi^*$. We find a rapid adiabatic population transfer from S_2 to S_1 ($\tau = 17 \pm 1$ fs) in our simulation. However, when the adiabatic states are decomposed into their diabatic components, and in particular into their $\pi\pi^*$ and $n\pi^*$ components, we see that the growth in the $n\pi^*$ character ($\tau = 37 \pm 9$ fs) is in close agreement with the time constant determined from the simulated spectrum ($\tau = 41 \pm 14$ fs). This shows that the 526 eV signal in the spectrum is due to the electronic $n\pi^*$ character.*

Comment: (4) The TRPE experiment of Suzuki and coworkers (Miura et al., Ref. 35) stands out by its excellent time resolution and excellent signal-to-noise ratio. It first definitively established the lifetime of the p π i* state of thymine. This work should be discussed in the introduction rather than on p. 9.

Response: We agree with the reviewer that this work deserves mention in the introduction. We have adjusted the paragraph discussing the experiments:

Experimental evidence has implicated the $n\pi^$ state in the early dynamics. Indeed, by determining the gas phase oxygen-edge time-resolved X-ray absorption spectrum, Wolf et al. [20] found a fast component ($\tau = 60 \pm 30$ fs) which was attributed to population of the $n\pi^*$ state. This was further corroborated in a recent time-resolved photoelectron spectrum reported by Miura et al. [29] ($\tau = 39 \pm 1$ fs). Thus, the wavepacket appears to already transfer some of its population to the $n\pi^*$ state within the first 100 fs. [...]*

Comment: (5) The term “wavepacket dynamics” is used in the abstract and throughout the manuscript. It should succinctly be explained whether 16 classical trajectories (of which 10 survive) with overall 67 Gaussian basis functions do accurately represent quantum wavepackets which are split at multiple conical intersections.

Response: We thank the reviewer for the comment and have adjusted the text. First, with respect to the method used, we now mention in the introduction wavepacket dynamics is performed with the *ab initio* multiple spawning method:

[...] Here, we apply the SCC with singles and doubles (EOM-SCCSD) method to simulate the first 100 fs after photoexcitation using *ab initio* multiple spawning (AIMS) [14, 15] [...]

With respect to the initial conditions and number of trajectory basis functions, we have made an adjustment when discussing the dissociative pathway in order to make the limited statistics clear:

In the dynamics simulation, only 2 out of 16 initial conditions (13%) lead to hydrogen dissociation at the N₁-H bond. Given this limited number of initial conditions, it is difficult to estimate how common this pathway is. Inspection of the natural transition orbitals (NTOs) of one of the conditions [...]

For the $\pi\pi^*$ to $n\pi^*$ conversion, the time constants are converged, see Figure S.4 in Supporting Information S1.

Comment: (6) Caption Fig. 2: The energy and time resolutions of transient absorption (TA) pump-probe (PP) spectra are Fourier limited. In gas-phase samples (no inhomogeneous broadening) the energy resolution is completely determined by the pulse durations. What is the justification for the Gaussian broadening in time and the additional Gaussian broadening of 0.3 eV in energy? These important details of the simulations should be explained in the Theoretical Methods section or in a section of the SI rather than in the figure caption.

Response: We thank the reviewer for bringing this to our attention. The Gaussian broadening in time is based on the time resolution in the experiment (70 fs pulse duration [20]). The degree of uncertainty can be inferred from the experimental spectrum, where the signal extends into negative times. The energy broadening was selected to produce a width consistent with the experiment. We now provide additional details in the Methods section.

Comment: (7) The non-augmented cc-pVTZ basis is inappropriate for the $\pi\sigma^*$ PES in the vicinity of the FC zone, where the σ^* orbital is very diffuse. The parts of the adiabatic PES which are of $\pi\sigma^*$ character likely are too high in energy. This limitation of the simulations should be discussed.

Response: We thank the reviewer for pointing out that we did not discuss the diffuse character of the state in the Franck-Condon region. We have performed some additional calculations which are discussed in the revised text:

Additional calculations support the predicted $\pi\sigma^$ pathway. Both of the dissociative initial conditions display identical behavior when the S_3 state is included in the dynamics simulation, showing that its inclusion does not suppress the channel (see Supporting Information S6). Moreover, while the $\pi\sigma^*$ state has Rydberg character in the Franck-Condon region, this character decreases as the N_1 -H bond extends and the state becomes involved in the dynamics. Potential energy curves along the N_1 -H bond with a diffuse basis set (aug-cc-pVDZ) and with a higher-order correlation treatment (coupled cluster with perturbative triples, CC3 [2]) suggest that the pathway is still present with more accurate treatments (see Supporting Information S7).*

Comment: (8) Having learned that the CCSD method describes the PES of thymine well, the reader also would like to know what went wrong in the previous simulations, e.g. Ref. 6. Is it the lack of dynamical correlation energy in the CASSCF method?

Response: This is a difficult question to answer and we have decided not to speculate in the main text. It seems that the $\pi\pi^*$ minimum obtained in CASSCF becomes shallower or disappears completely as dynamical correlation is included in the electronic structure description (as found with ADC(2), DFT, and now also with EOM-CCSD), but we think it is an open question whether there is a direct causal link.

References

- [1] Schneider, M. *et al.* Photodissociation of thymine. *Phys. Chem. Chem. Phys.* **8**, 3017–3021 (2002).
- [2] Koch, H., Christiansen, O., Jørgensen, P., Sanchez de Merás, A. M. & Helgaker, T. The CC3 model: An iterative coupled cluster approach including connected triples. *J. Chem. Phys.* **106**, 1808–1818 (1997).
- [3] Kjøenstad, E. F., Myhre, R. H., Martínez, T. J. & Koch, H. Crossing conditions in coupled cluster theory. *J. Chem. Phys.* **147**, 164105 (2017).
- [4] Kjøenstad, E. F. & Koch, H. Resolving the notorious case of conical intersections for coupled cluster dynamics. *J. Phys. Chem. Lett.* **8**, 4801–4807 (2017).
- [5] Kjøenstad, E. F. & Koch, H. An orbital invariant similarity constrained coupled cluster model. *J. Chem. Theory Comput.* **15**, 5386–5397 (2019).
- [6] Schnack-Petersen, A. K., Koch, H., Coriani, S. & Kjøenstad, E. F. Efficient implementation of molecular CCSD gradients with Cholesky-decomposed electron repulsion integrals. *J. Chem. Phys.* **156**, 244111 (2022).
- [7] Kjøenstad, E. F. & Koch, H. Communication: Non-adiabatic derivative coupling elements for the coupled cluster singles and doubles model. *J. Chem. Phys.* **158**, 161106 (2023).

- [8] Kjøenstad, E. F., Angelico, S. & Koch, H. Coupled cluster theory for nonadiabatic dynamics: nuclear gradients and nonadiabatic couplings in similarity constrained coupled cluster theory. *To be submitted.* (2024).
- [9] Purvis III, G. D. & Bartlett, R. J. A full coupled-cluster singles and doubles model: The inclusion of disconnected triples. *J. Chem. Phys.* **76**, 1910–1918 (1982).
- [10] Stanton, J. F. & Bartlett, R. J. The equation of motion coupled-cluster method. A systematic biorthogonal approach to molecular excitation energies, transition probabilities, and excited state properties. *The Journal of Chemical Physics* **98**, 7029–7039 (1993).
- [11] Hättig, C. Structure optimizations for excited states with correlated second-order methods: CC2 and ADC(2). *Adv. Quantum Chem.* **50**, 37–60 (2005).
- [12] Köhn, A. & Tajti, A. Can coupled-cluster theory treat conical intersections? *J. Chem. Phys.* **127**, 044105 (2007).
- [13] Williams, D. M. G., Kjøenstad, E. F. & Martínez, T. J. Geometric phase in coupled cluster theory. *J. Chem. Phys.* **158**, 214122 (2023).
- [14] Ben-Nun, M., Quenneville, J. & Martínez, T. J. Ab initio multiple spawning: Photochemistry from first principles quantum molecular dynamics. *J. Phys. Chem. A* **104**, 5161–5175 (2000).
- [15] Ben-Nun, M. & Martínez, T. J. Ab initio quantum molecular dynamics. *Adv. Chem. Phys.* **121**, 439–512 (2002).
- [16] Stojanović, L. *et al.* New insights into the state trapping of UV-excited thymine. *Molecules* **21**, 1603 (2016).
- [17] Wolf, T. J. A. *et al.* Observation of ultrafast intersystem crossing in thymine by extreme ultraviolet time-resolved photoelectron spectroscopy. *J. Phys. Chem. A* **123**, 6897–6903 (2019).
- [18] Wolf, T. J. A. & Gühr, M. Photochemical pathways in nucleobases measured with an X-ray FEL. *Philos. Trans. R. Soc. A* **377**, 20170473 (2019).
- [19] Park, W., Lee, S., Huix-Rotllant, M., Filatov, M. & Choi, C. H. Impact of the dynamic electron correlation on the unusually long excited-state lifetime of thymine. *J. Phys. Chem. Lett.* **12**, 4339–4346 (2021).
- [20] Wolf, T. J. A. *et al.* Probing ultrafast $\pi\pi^*/n\pi^*$ internal conversion in organic chromophores via K-edge resonant absorption. *Nat. Commun.* **8**, 29 (2017).
- [21] Crespo-Hernandez, C. E., Cohen, B., Hare, P. M. & Kohler, B. Ultrafast excited-state dynamics in nucleic acid. *Chem. Rev.* **104**, 1977–2020 (2004).

- [22] Canuel, C. *et al.* Excited states dynamics of DNA and RNA bases: Characterization of a stepwise deactivation pathway in the gas phase. *J. Chem. Phys.* **122**, 074316 (2005).
- [23] Ullrich, S., Schultz, T., Zgierski, M. Z. & Stolow, A. Electronic relaxation dynamics in DNA and RNA bases studied by time-resolved photoelectron spectroscopy. *Phys. Chem. Chem. Phys.* **6**, 2796–2801 (2004).
- [24] McFarland, B. K. *et al.* Ultrafast x-ray auger probing of photoexcited molecular dynamics. *Nat. Commun.* **5**, 4235 (2014).
- [25] Kang, H., Lee, K. T., Jung, B., Ko, Y. J. & Kim, S. K. Intrinsic lifetimes of the excited state of DNA and RNA bases. *J. Am. Chem. Soc.* **124**, 12958–12959 (2002).
- [26] Asturiol, D., Lasorne, B., Robb, M. A. & Blancafort, L. Photophysics of the π , π^* and n , π^* states of thymine: MS-CASPT2 minimum-energy paths and CASSCF on-the-fly dynamics. *J. Phys. Chem. A* **113**, 10211–10218 (2009).
- [27] Nix, M. G. D., Devine, A. L., Cronin, B. & Ashfold, M. N. R. Ultraviolet photolysis of adenine: Dissociation via the $^1\pi\sigma^*$ state. *J. Chem. Phys.* **126**, 124312 (2007).
- [28] Roberts, G. M. & Stavros, V. G. The role of $\pi\sigma^*$ states in the photochemistry of heteroaromatic biomolecules and their subunits: insights from gas-phase femtosecond spectroscopy. *Chem. Sci.* **5**, 1698–1722 (2014).
- [29] Miura, Y. *et al.* Formation of long-lived dark states during electronic relaxation of pyrimidine nucleobases studied using extreme ultraviolet time-resolved photoelectron spectroscopy. *J. Am. Chem. Soc.* **145**, 3369–3381 (2023).

Response to reviewer's comments for "Unexpected hydrogen dissociation in thymine predicted by coupled cluster theory"

Eirik F. Kjørnstad^{*1,2,3}, O. Jonathan Fajen^{1,2}, Alexander C. Paul³,
Sara Angelico³, Dennis Mayer⁴, Markus Gühr^{4,5},
Thomas J. A. Wolf¹, Todd J. Martínez^{*1,2}, Henrik Koch^{*3}

¹Department of Chemistry, Stanford University, Stanford, CA, USA.

²Stanford PULSE Institute, SLAC National Accelerator Laboratory,
Menlo Park, CA, USA.

³Department of Chemistry, Norwegian University of Science and
Technology, Trondheim, 7491, Norway.

⁴Deutsches Elektronen-Synchrotron DESY, Hamburg, Germany.

⁵Institute of Physical Chemistry, University of Hamburg, Hamburg,
Germany.

Contributing authors: eirik.kjonstad@ntnu.no;
todd.martinez@stanford.edu; henrik.koch@ntnu.no;

We thank all the reviewers for their comments, which we address point-by-point below and which we believe have led to significant improvements of the manuscript.

Reviewer 1

Comment: The authors have submitted a thoroughly revised version of their manuscript and the criticism of the reviewers has been discussed in great detail in

their rebuttal (and is echoed in their revised manuscript). I have only a few minor comments, otherwise I can clearly recommend this work for publication.

Response: We thank the reviewer for carefully reading the manuscript and we appreciate the useful comments provided.

Comment: Concerning the last comment (#8) of reviewer 4, I feel that this is a legitimate (and in way also central) question, which boils down to "can we understand why the results of the present study are different (and better) than previous work" (where the pure "coincidence with experiment" is maybe not the most convincing argument)? As there is an overlap of the present author team to the authors of one of the earlier studies, which employs rather orthogonal techniques like CASSCF (and MS-CASPT2), I could expect that a more detailed answer could be given. It would be (as I think) also worthwhile to point out that findings like the pi-sigma* channel are more difficult in a CASSCF-based approach, as the active space choice may exclude certain channels right from the outset.

Response: When the CCSD ground state is well described, EOM-CCSD provides an accurate balanced description of the multiconfigurational effects in the excited states, when these are dominated by single excitations. This is indeed the case for the region we investigate in this paper. Therefore, there is every reason to believe that the CCSD description is accurate and more so than CASSCF which does not capture dynamical correlation and can depend on the choice of active space.

Comment: Just as a comment that does not require further action by the authors: Fig. S20 shows that the pi sigma* pathway is a stable feature (and that is what the authors also state in the main text), but it also clearly shows that the basis set limit is far away and that the prediction of excited state dynamics (in particularly at higher energies with increasing density of states) remains an adventure.

Response: We thank the reviewer for the comment, and we agree that there are still open challenges to the application of the method for systems with a high density of states, as is also the case for other electronic structure methods.

Comment:

- Fig. 3: In A and C, the units of g and h are not defined (I assume unitless normal coordinates, but it would be better, if this is stated). For panels B and D, I suggest to convert the abscissa to eV (to match with other graphs).
- Fig. 5: In principle, I am OK with the shaded area, but there is one thing that creates confusion. The experimental spectrum is labelled with "Experiment" and a dotted line connects the label with the graph of the experimental spectrum, coming from outside the shaded area and ending in this area. This (at least for me) gives at first glance the impression that the label "Experiment" refers to the shaded area. I suggest to move this label to a different position and to also explicitly label the shaded area with, e.g., "no TKER data" or something similar.

- Again Fig. 5 (and text on page 7): I assume that Q is expressed in unitless (=mass and frequency weighted) coordinates. This might be added at an appropriate place.
- I also noted that the authors partially use "au" as "atomic units" (mainly in the technical part and in the SI), but at the same time in Fig. 5 "a.u." stands for arbitrary units (absorption). I am normally a big fan of spelling out atomic units, so the atomic unit of time, for instance, is written as \hbar/E_h . This may also be helpful to reader to understand the translation to femtoseconds.
- I also noted that in some instances in the SI, the atomic unit of time is used, where the reference to the content of the graphic is only indirect, as there fs are used as units, see Figs. S14 and S15.
- In Section S9 "au" is used for lengths (which is confusing, as "tau" used as the symbol is often associated with time), I suggest to use a_0 here (the official symbol for Bohr units).

Response: We thank the reviewer for the useful suggestions, which we have now incorporated in both the figures and the text.

Reviewer 2

Comment: In their revised manuscript, the authors have greatly improved the manuscript and solved the issues raised by the reviewers. The manuscript can now in principle be published in Nature Communications. I have only a few smaller comments, as given below:

Response: We thank the reviewer for acknowledging the improvement of the manuscript.

Comment: 1) Despite it being many items, I would recommend the authors to reference all SI items in the main text, and not only the sections. I believe that this is actually Nature Communications policy, so the authors might need to add such references at some point anyways

Response: We have now referenced all SI items in the Methods section.

Comment: 2) Based on the replies in the rebuttal, I have the impression that the authors consider employing the new electronic structure method as one of the main messages of the manuscript. In that case, I strongly suggest to include this message in the abstract, where it is currently missing.

Response: We have now modified the abstract to include this message.

Comment: 3) I suggest to add to the statement of "We find no significant pipi^* trapping and no direct pipi^* relaxation to the ground state." on page 4 the time scale of 100fs, to avoid any possible misunderstandings in the readers. A reminder on

the limitations of the level of theory is found elsewhere, but could be added in this sentence as well.

Response: We have now added the statement suggested by the reviewer on page 4.

Comment: 4) I still do not fully understand with the several general statements claiming the "highest electronic structure level of theory". As I stated before, I believe that one cannot simply line up all levels of theory and define a clear "highest". At the very least, the authors should modify the claim to "highest single-reference level of theory", which is a statement with which I could agree.

Response: We have now modified the statements as suggested by the reviewer.

Unexpected hydrogen dissociation in thymine: predictions from a novel coupled cluster theory

Henrik Koch, Eirik Kjønstad, Otto Jonathan Fajen, Alexander Paul, Sara Angelico, Dennis Mayer, Markus Gühr, Thomas Wolf, and Todd Martinez

The manuscript reports a theoretical study of the ultrafast dynamics of isolated thymine molecules following photoexcitation at UV energies, using wavepacket dynamics together with a new coupled cluster method. The excited state dynamics of isolated thymine molecules have been studied previously, but I am willing to accept the authors assertion that this study has been undertaken at the highest level of theory to date. The study identifies that initial photoexcitation is to the 'bright' S_2 state, which arises via a $\pi^* \leftarrow \pi$ promotion. The S_1 state has predominant $n\pi^*$ character in the vertical region, but photoexcited molecules rapidly sample the region of conical intersection (CI) between the S_2 and S_1 states and most of the initial S_2 population transfers to the S_1 state at this CI. This photophysical interpretation, and the timescale predicted by the present simulation are broadly consistent with the results of prior work. The predicted time-resolved X-ray absorption spectrum of thymine following ultrafast UV photoexcitation, measured at the oxygen-edge, also matches a prior experimental measurement; the signature of the emerging and (over a short timescale) persistent $n\pi^*$ population is clearly revealed. The results described thus far provide little fundamental new knowledge, but certainly reinforce and validate prior discussions. However, the present calculations also reveal an additional (minor) decay pathway, involving H atom loss via breaking of the N_1 -H bond. Activity in such a channel might not be viewed as surprising, at least by those familiar with other the photophysics of other N-containing heterocycles, but the present work provides direct evidence that it may play a role.

The submitted manuscript is very readable, demonstrates use of a very recent and improved computational method for studying excited state photophysics in medium sized (*i.e.* ~ 10 (light) atom) molecules, confirms much of the current early-time photochemical knowledge for this molecule and identifies some activity in a hitherto unreported dissociation pathway. As such, I'm content that it passes the suitability threshold for publication in *Nature Communications*. I now offer suggestions as to how (in my opinion) the manuscript could be improved, but the authors should feel free to decide which (if any) they choose to act on.

In several places I felt that the authors were trying a bit too hard to 'sell' their work. The title provides a good case in point. Hydrogen does not 'dissociate' – it's an N–H bond that dissociates. (The phrase hydrogen dissociation also appears elsewhere in the manuscript). I would also quibble with the use of the word 'Unexpected'. 'Hitherto unreported' might be fairer. As the authors discuss later in the manuscript, analogy with other N-heterocycles implies that there will be (more than) one dissociative $(n/\pi)\sigma^*$ potential that could support N–H bond fission. The process has not been reported hitherto in thymine but, again, the authors rightly point out that this may just be because the right type of experiment has yet been performed at suitably high excitation energies. I'm also not fond of claims to be the 'first' to do something – as claimed twice (for essentially the same point) in the paragraph leading into Fig 1 and reiterated again in the concluding paragraph.

Generally interested readers might welcome a bit more preamble. The Introductory paragraph starts with 'Thymine, like other nucleobases, undergoes ultrafast radiationless relaxation back to the ground state after being excited by ultraviolet radiation. This property has been tied to the resilience of genetic material against light-induced damage [1]. However, the exact mechanism of this decay is not fully understood and has been a subject of debate for several decades' – all of which is true.

Most readers (*i.e.* those outside the ultrafast, first 100 fs, isolated molecule community), however, might then be disappointed to find that the ‘new insights’ reported here never actually get to the ‘back to the ground state’ bit. For general interest, it might be worth being clearer that this work only addresses the *earliest* time dynamics in *isolated* thymine molecules. It is silent on how excited thymine molecules in the $n\pi^*$ state couple back to the ground state, or how they would decompose further in the absence of collisions. The ‘resilience of genetic material against light-induced damage’ applies when molecules such as thymine are in a condensed phase environment, and the energy introduced by photon absorption can be dissipated to the background.

Readers might also appreciate a bit more clarity about the relevant excitation energies/wavelengths. The Wolf *et al* results (ref 16) against which the present theoretical data are compared were obtained following excitation with a UV pulse centred at a wavelength of 267 nm (*i.e.* the red wing of the parent absorption spectrum). I did not find an explicit statement in the present manuscript as to the excitation energy used in the simulations. (I may have missed it). But, from Fig. 5, I assume it was the energy appropriate for vertical excitation to the *calculated* S_2 state – *i.e.* a photon energy ~ 0.5 eV higher than used in the Wolf *et al.* experiments. This discrepancy raises several minor worries in my mind. Should one be surprised that there is still a >0.2 eV mismatch between the maxima in the long wavelength feature in the experimental and computed spectra of thymine, given the assertion that the paper reports ‘the highest level of electronic structure theory performed on a molecular system of this size’? The dynamical simulations return decay lifetimes and channel branching ratios based on the calculated potentials and assumed excitation energy. The 267 nm experiments presumably sample the S_2 potential at a slightly different starting geometry (away from vertical) and at a different total energy. To what extent should one expect the measured and predicted timescales to be the same? The manuscript currently emphasises the good agreement, but nowhere discusses the extent to which the two studies are (or are not) exploring the same thing. The calculated branching ratios are reported to the nearest percent, which is fair enough; that’s what the simulations return. But as Fig 5 shows, the calculations still don’t get the S_2 state energy correct. Do the authors have a feel for how the estimated branching into the N-H bond fission channel depends on the reliability of their $\pi\pi^*$ and $\pi\sigma^*$ excitation energies along Q in Fig 4B (the cut along the $\pi\sigma^*$ potential at large Q looks very questionable) or to the excitation energy assumed? In similar vein, the current manuscript contains the sentence on ‘Other experiments have appeared to disconfirm the existence of a dissociative $\pi\sigma^*$ channel’ but gives no reference(s) indicating to which other experiments the statement applies.

The last sentence in the caption to Fig. 5 ‘The shaded area corresponds to wavelengths where TKER spectra have not been recorded’ is rather misleading. TKER spectra have not been recorded at any shorter wavelengths on the right-hand side of that figure either. More sensible might be to use a shaded region to show where TKER measurements *have* been reported.

I have not checked the references, but it is obvious that refs 16, 18, 19, 27, 43 and 45 are all missing key details.